# CAMBRIAN-*S*: TOWARDS SPATIAL SUPERSENSING IN VIDEO

**Shusheng Yang**[1,*]**, Jihan Yang**[1,*]**, Pinzhi Huang**[1,†]**, Ellis Brown**[1,†]**, Zihao Yang**[1]**,**
**Yue Yu**[1]**, Shengbang Tong**[1]**, Zihan Zheng**[1]**, Yifan Xu**[1]**, Muhan Wang**[1]**,**
**Rob Fergus**[1]**, Yann LeCun**[1]**, Li Fei-Fei**[2]**, Saining Xie**[1]

[*]Equal contribution, [†]Core contributor, [1]New York University, [2]Stanford University

## ABSTRACT

We argue that progress in true multimodal intelligence calls for a shift from reactive, task-driven systems and brute-force long context towards a broader paradigm of *supersensing*. We frame spatial supersensing as four stages beyond linguistic-only understanding: semantic perception (naming what is seen), streaming event cognition (maintaining memory across continuous experiences), implicit 3D spatial cognition (inferring the world behind pixels), and predictive world modeling (creating internal models that filter and organize information). Current benchmarks largely test only the early stages, offering narrow coverage of spatial cognition and rarely challenging models in ways that require true world modeling. To drive progress in spatial supersensing, we present VSI-SUPER, a two-part benchmark: VSR (long-horizon visual spatial recall) and VSC (continual visual spatial counting). These tasks require arbitrarily long video inputs yet are resistant to brute-force context expansion. We then test data scaling limits by curating VSI-590K and training Cambrian-*S*, achieving $+30\%$ absolute improvement on VSI-Bench without sacrificing general capabilities. Yet performance on VSI-SUPER remains limited, indicating that scale alone is insufficient for spatial supersensing. We propose *predictive sensing* as a path forward, presenting a proof-of-concept in which a self-supervised next-latent-frame predictor leverages *surprise* (prediction error) to drive memory and event segmentation. On VSI-SUPER, this approach substantially outperforms leading proprietary baselines, showing that spatial supersensing requires models that not only see but also anticipate, select, and organize experience.

## 1 INTRODUCTION

A video is not just a sequence of frames in isolation. It is a continual, high-bandwidth projection of a hidden, evolving 3D world onto pixels (Gibson, 2014; Marr, 2010). Although multimodal large language models (MLLMs) have advanced rapidly by pairing strong image encoders with language models (Achiam et al., 2023; Team et al., 2024; Anthropic, 2024; Liu et al., 2023; Tong et al., 2024a), most video extensions (Wang et al., 2024d; Li et al., 2025a; Bai et al., 2025a) remain fundamentally constrained. They still treat video as sparse frames, underrepresent spatial structure and dynamics (Yang et al., 2024d), and lean heavily on textual recall (Zohar et al., 2025), thus overlooking what makes the video modality uniquely powerful.

In this paper, we argue that advancing toward true multimodal intelligence requires a shift from language-centric perception toward spatial *supersensing:* the capacity not only to see, but also to construct, update and predict with an implicit model of the 3D world from continual sensory experience. We do not claim to realize supersensing here; rather, we take an initial step toward it by articulating the potential path and by demonstrating early prototypes along that path:

0. **(Linguistic-only understanding):** no sensory capabilities; reasoning confined to text and symbols. Current MLLMs have progressed beyond this stage, yet still retain traces of its bias.
1. **Semantic perception:** parsing pixels into objects, attributes, and relations. This corresponds to the strong multimodal *"show and tell"* capabilities present in MLLMs.
2. **Streaming event cognition:** processing live, unbounded streams while proactively interpreting and responding to ongoing events. This aligns with efforts to make MLLMs real-time assistants.
3. **Implicit 3D spatial cognition:** understanding video as projections of a 3D world. Agents must know where and how things relate, and how configurations change over time. Today's video models remain limited here.
4. **Predictive world modeling:** the brain makes *unconscious inferences* (Von Helmholtz, 1867) by predicting latent world states based on prior expectations. When these predictions are violated, surprise guides attention, memory, and learning (Friston, 2010; Stahl & Feigenson, 2015; Kennedy et al., 2024). However, current

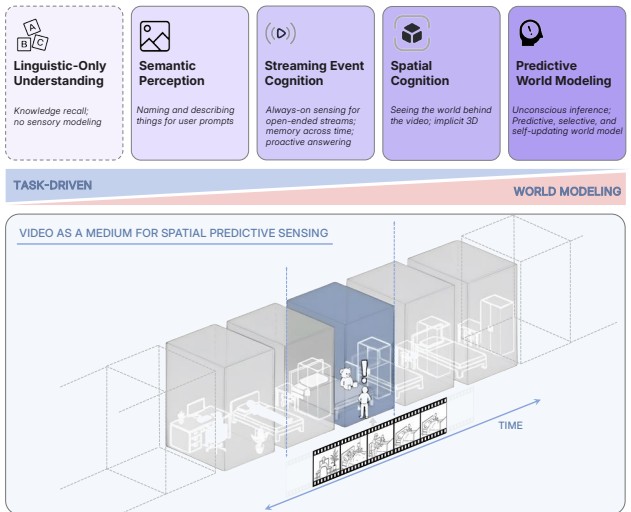

Figure 1: **From pixels to predictive mind.** We look beyond *linguistic-only* understanding to envision multimodal intelligence that sees, remembers, and reasons as part of a continuous, lived world. It begins with *semantic perception*: naming and describing what is seen. *Streaming event cognition* goes further, enabling always-on sensing across continuous input streams. *Spatial cognition* captures the implicit 3D structure of video. Finally, a *predictive world model* emerges, one that learns passively from experience, updates through prediction and surprise, and retains information for future use. **Lower illustration:** Video serves as ideal experimental domain. Models must advance from frame-level Q&A to constructing implicit world models that enable deeper spatial reasoning, scale to unbounded horizons, and achieve supersensing.

multimodal systems lack an internal model that anticipates future states and uses surprise to organize perception for memory and decision making.

Our paper unfolds in three parts. **First** (§ 2), we re-examine existing benchmarks through the lens of our supersensing hierarchy. We find that most benchmarks map to the first few stages, while some, such as VSI-Bench (Yang et al., 2024d), begin to probe spatial reasoning. However, none sufficiently address the final crucial stage of predictive world modeling. To make this gap concrete and motivate a shift in approach, we introduce VSI-SUPER (VSI stands for *visual-spatial intelligence*), a two-part benchmark for spatial supersensing: VSI-SUPER Recall (VSR) targets long-horizon spatial observation and recall, while VSI-SUPER Count (VSC) tests continual counting across changing viewpoints and scenes. Built from arbitrarily long spatiotemporal videos, these tasks are deliberately resistant to the predominant multimodal recipe; they require perception to be *selective* and *structured* rather than indiscriminately accumulated. Even the best long-context commercial models fail on it.

**Second** (§ 3), we investigate whether spatial supersensing is simply a data problem. We curate *VSI-590K*, a spatially focused instruction-tuning corpus over images and videos, which we use to train *Cambrian-S*, a family of spatially-grounded video MLLMs. Under the current paradigm, careful data design and training push Cambrian-*S* to state-of-the-art spatial cognition on VSI-Bench (>30% absolute gain) without sacrificing general capabilities. Nevertheless, Cambrian-*S* still falls short on VSI-SUPER, indicating that scaling alone is not sufficient for spatial supersensing.

This motivates the **third** and final part (§ 4), where we propose *predictive sensing* as a first step toward a new paradigm. We present a proof-of-concept solution built upon self-supervised next-latent-frame prediction. Here, we leverage the model's prediction error, or "surprise," for two key functions: (1) managing memory by allocating resources to unexpected events, and (2) event segmentation, breaking unbounded streams into meaningful chunks. We demonstrate that this approach, though simple, significantly outperforms strong long-context baselines such as Gemini-2.5 on our two new tasks. Although not a final solution, this result provides compelling evidence that the path to true supersensing requires models that not only *see* but actively predict and learn from the world.

Our work makes the following contributions. (**1**) We define a hierarchy for spatial supersensing and introduce VSI-SUPER, a supersensing benchmark that reveals the limitations of the current paradigm. (**2**) We develop Cambrian-*S*, a state-of-the-art model that pushes the limits of spatial cognition. Cambrian-*S* serves as a powerful new baseline, and, by delimiting the boundaries of current methods on our new benchmark, paves the path for a new paradigm. (**3**) We propose predictive sensing as a

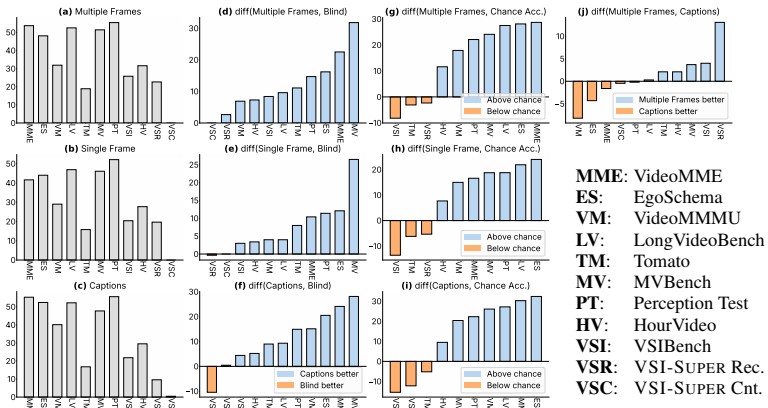

**Figure 2: Benchmark diagnostic results reveal varying dependence on visual input.** We evaluate model under three conditions: (**a**) multiple frames, (**b**) single frame, and (**c**) frame captions, benchmarked against chance-level and blind test results. VSR and VSC are new supersensing benchmarks in Section 2.2.

promising new direction for MLLMs, showing that leveraging model surprise is more effective for long-horizon spatial reasoning than passive context expansion.

## 2 BENCHMARKING SPATIAL SUPERSENSING

To ground our pursuit of spatial supersensing, we first establish how to measure it. This section undertakes a two-part investigation into benchmarking this capability. We begin by auditing a suite of popular video MLLM benchmarks, where our analysis (Figs. 2 and 12) reveals that they overwhelmingly focus on linguistic understanding and semantic perception while neglecting the more advanced spatial and temporal reasoning required for supersensing (Section 2.1). To address this critical gap, we then introduce VSI-SUPER, a new benchmark specifically designed to probe these harder, continual aspects of spatial intelligence in arbitrarily long streaming scenarios (Section 2.2). We use this benchmark to test the limits of the current MLLM paradigm.

### 2.1 DECONSTRUCTING EXISTING VIDEO BENCHMARKS

To assess if existing benchmarks evaluate *true visual sensing* or simply rely on language priors, we conduct a series of diagnostic tests. We use our base Cambrian-1 (Tong et al., 2024a) model to probe a suite of representative video benchmarks under varied input conditions, allowing us to disentangle the underlying task demands from the capabilities of more complex video-specific architectures. We establish several experimental conditions for feeding video inputs: **Multiple Frames** (32 uniformly sampled frames), **Single Frame** (the middle frame), and **Frame Captions**. We compare these against two baselines: **Blind Test** (all visual input is ignored) and **Chance Acc** (randomly guessing). We conduct a fine-grained analysis of each benchmark's characteristics by comparing performance across these conditions and baselines (`diff(A,B) = A-B`).

Results presented in Fig. 2 (a-c) demonstrate that, an image-based MLLM without any video post-training, can attain reasonable performance across many benchmarks, in some instances surpassing chance-level accuracy by 10-30% (see Fig. 2-g,h). This suggests that much of the knowledge these benchmarks target is accessible via standard single-image instruction-tuning pipelines. Nevertheless, on two existing datasets, VSI-Bench (Yang et al., 2024d) and Tomato (Shangguan et al., 2025), the model's performance falls below chance-level. Employing textual captions in place of visual inputs also yields notable performance improvements, surpassing chance accuracy by more than 20% on benchmarks such as EgoSchema (Mangalam et al., 2023), VideoMME (Fu et al., 2025), LongVideoBench (Wu et al., 2024b), VideoMMMU (Hu et al., 2025a), Perception Test (Patraucean et al., 2023), and MVBench (Li et al., 2024d) (Fig. 2-i). Similar conclusions can be drawn when comparing benchmark performance against blind test results (Fig. 2-d,f). Such performance implies that these benchmarks primarily probe abilities inferable from textual summaries of video content. The performance difference between using "multiple frames" and "frame captions" (Fig. 2-j) indicates a benchmark's demand for nuanced visual sensing or language-centric nature. Our analysis places VideoMMMU, EgoSchema, VideoMME, Perception Test, and LongVideoBench in this latter category, indicating their potential reliance on *linguistic understanding* rather than visual cues.

> 🔖 Existing benchmarks overwhelmingly focus on linguistic understanding and semantic perception while neglecting spatial and temporal reasoning required for supersensing.

We hope to emphasize the inherent challenges in benchmarking and the impracticality of creating a single, all-encompassing benchmark to evaluate every capability. For example, reliance on language priors should not be viewed merely as a drawback, as access to rich world knowledge and its effective retrieval is undoubtedly beneficial in many scenarios. We argue that **video benchmarks should not be treated as measuring a single, uniform notion of "video understanding." Instead, their design and evaluation should be grounded in the specific capabilities they aim to assess.** The preceding analyses are therefore intended to guide the development of tasks that more effectively drive progress towards *spatial supersensing*, which will be the central focus of the rest of the paper.

## 2.2 VSI-SUPER: TOWARDS BENCHMARKING SPATIAL SUPERSENSING IN MLLMS

Referring to Fig. 1, spatial supersensing requires MLLMs to have four key capabilities. However, as outlined by our analysis in Fig. 2, most existing video QA benchmarks mainly evaluate the linguistic understanding and semantic perception aspects, which are more reactive and driven by specific tasks (Fu et al., 2025; Mangalam et al., 2023; Hu et al., 2025a). While recent research has begun to address streaming event cognition through continual sensing, memory architectures, and proactive answering (Chen et al., 2024c; Qian et al., 2025; Niu et al., 2025; Wu et al., 2024a; Song et al., 2024; Zhang et al., 2025a), this capability is often engineered at test time rather than being a native model skill. Additionally, although VSI-Bench (Yang et al., 2024d) takes an initial step toward examining spatial cognition, its videos remain short-form and single-scene, and it neither formalizes the problem nor evaluates the essential capability of predictive modeling of the world.

To illuminate the gap between current MLLMs and spatial supersensing, we introduce VSI-SUPER, a two-part benchmark for continual spatial sensing. The tasks are intuitive and generally easy for humans, where one simply watches and keeps track of what happens, but they remain surprisingly challenging for machines. They demand selective filtering and structured accumulation of visual information across unbounded spatial videos to maintain coherent understanding and answer questions. Importantly, they are resistant to brute-force context expansion, exposing the need for true spatial reasoning. We detail the two components below.

**VSI-SUPER Recall: Long-horizon spatial observation and recall.** The VSR benchmark requires MLLMs to observe long-horizon spatiotemporal videos, and sequentially recall the locations of an unusual object. As shown in Fig. 3, to construct this benchmark, human annotators use an image editing model (*i.e.*, Gemini (Comanici et al., 2025)) to insert surprising or out-of-place objects (*e.g.*, a Teddy Bear) into four distinct frames (and spatial location) of a

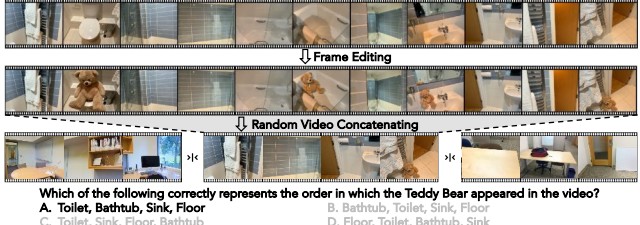

Figure 3: **VSR Task**: recall the placements of novel objects in the correct order of their appearance across arbitrarily long videos.

video capturing a walkthrough of an indoor environment (Dai et al., 2017; Yeshwanth et al., 2023; Baruch et al., 2021). This edited video is then concatenated with other similar room-tour videos to create an arbitrarily long and continuous visual stream. This task parallels the needle-in-a-haystack (NIAH) test commonly used in the language domain to stress test the long-context capabilities of LLMs (Liu et al., 2024b). Similar NIAH setups have also been proposed for long-video evaluation (Zhao et al., 2024; Wei et al., 2025; Hu et al., 2025b). However, unlike benchmarks that insert unrelated text segments or frames, VSR preserves the realism of the "needle" through in-frame editing. It further extends the challenge by requiring sequential recall, effectively a multi-hop reasoning task, and remains arbitrarily scalable in video length. To thoroughly evaluate model performance across different time scales, the benchmark is provided in five durations: 10, 30, 60, 120, and 240 minutes. Further details on the VSR benchmark construction are provided in Appendix D.

**VSI-SUPER Count: Continual counting under changing viewpoints and scenes.** Here we test the capacity of MLLMs to continuously accumulate information in long-form spatial videos. To build VSC, we concatenate multiple room-tour video clips from VSI-Bench (Yang et al., 2024d) and task models with counting the *total* number of target objects across all rooms (see Fig. 4). This setting is challenging because the model must handle viewpoint shifts, repeat sightings, and scene

transitions, all while maintaining a consistent cumulative count. For humans, counting is an intuitive and generalizable process. Once the concept of "one" is understood, extending it to larger quantities is natural. In contrast, as we later demonstrate, current MLLMs lack true spatial cognition and depend excessively on learned statistical patterns.

In addition to standard evaluations (*i.e.*, ask question at the end of video), we query the model at multiple timestamps to assess its performance in streaming settings, where the correct answer in VSC evolves dynamically over time. To examine long-term consistency, VSC includes four video durations: 10, 30, 60, and 120 minutes. For this quantitative task, we report results using the mean relative accuracy ($\mathcal{MRA}$) metric, consistent with the VSI-Bench evaluation protocol (Yang et al., 2024d).

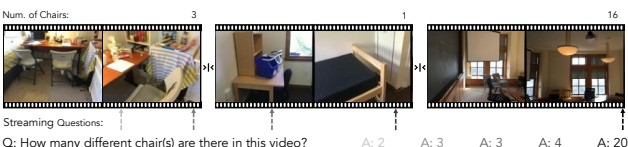

Figure 4: **Overview of the VSC**: count the total number of target objects across all rooms in long-horizon, multi-room videos.

**State-of-the-art models struggle on VSI-SUPER.** To test whether VSI-SUPER poses a real challenge for frontier MLLMs, we evaluate the latest Gemini-2.5-Flash (Team et al., 2024). As shown in Table 1, the model reaches its context limit when handling two-hour videos, despite a context length of 1,048,576 tokens. This highlights the *open-ended* nature of video understanding, where continuous streams effectively require an "infinite-in, infinite-out" context and can grow arbitrarily long, suggesting that simply scaling up tokens, context length, or model size may not suffice. Though synthetic, our benchmark reflects a real challenge in spatial supersensing: humans effortlessly integrate and retain information from ongoing sensory experiences that unfold over hours or years, yet current models lack comparable mechanisms for sustained perception and memory. Gemini-2.5-Flash demonstrates strong performance on semantic-perception and linguistic-understanding-focused video benchmarks such as VideoMME (Fu et al., 2025) and VideoMMMU (Hu et al., 2025a), achieving around 80% accuracy. However, even for 60-minute videos in VSI-SUPER that fall well within its context window, performance on VSR and VSC remains limited—only 41.5 and 10.9, respectively. As shown in Fig. 13, the model's predicted object counts fail to scale with video length or the true number of objects, instead saturating at a small constant value, suggesting a lack of generalization in counting ability and a reliance on training distribution priors.

| Model | VideoMME(Fu et al., 2025) | VideoMMMU(Hu et al., 2025a) | VSI-Bench(Yang et al., 2024d) | VSR | | VSC | |
| | | | | 60 min | 120 min | 60 min | 120 min |
|---|---|---|---|---|---|---|---|
| Gemini-2.5-Flash | 81.5 | 79.2 | 45.7 | 41.5 | Out of Ctx. | 10.9 | Out of Ctx. |

Table 1: **Gemini-2.5-Flash demonstrates strong performance on general video benchmarks but shows clear limitations towards spatial supersensing.**

**VSI-SUPER challenges current paradigm.** Although the task setup is simple, the challenge posed by VSI-SUPER goes beyond spatial reasoning and reveals fundamental limitations of current paradigm.

> 🔖 VSI-SUPER tasks challenge the belief that scaling alone guarantees progress.

By allowing arbitrarily long video inputs that emulate the dynamics of streaming cognition, VSI-SUPER is intentionally constructed to exceed any fixed context window. This design suggests that frame-by-frame tokenization and processing are unlikely to be computationally viable as a long-term solution. Humans address such problems efficiently and adaptively by selectively attending to and retaining only a small fraction of sensory input[1], often unconsciously (Fei-Fei et al., 2007; Von Helmholtz, 1867). This predictive and selective mechanism, core to human cognition, remains absent in current MLLMs but is fundamental to a predictive world model.

> 🔖 VSI-SUPER tasks demand generalization to new temporal and spatial scales at test time.

For example, VSC requires counting in arbitrarily long videos, similar to how humans, who understand the concept of counting, can extend it to any number. The key is not maintaining an extremely long context window, humans do not retain every visual detail from extended visual experiences, but rather learning the process of counting itself. Together, these challenges call for a paradigm shift. Rather than relying solely on scaling data, parameters, or context length, future models should learn internal world models, perceiving and predicting within an endless visual world across space and time.

---

[1]Each eye's 6 million cone photoreceptors can send about 1.6 Gbits/s, yet the brain uses only 10 bits/s to guide behavior (Koch et al., 2006; Zheng & Meister, 2025).

# 3 SPATIAL SENSING UNDER THE CURRENT PARADIGM

In this section, we try to answer the question: *Is limited spatial sensing simply a data issue?* We begin by enhancing Cambrian-1 (Tong et al., 2024a) with a series of architectural and training improvements to establish a stronger image MLLM as our base model (Section 3.1). Then, we construct a large-scale, spatial-focused instruction-tuning dataset, VSI-590K (Section 3.2) to provide a strong data foundation for spatial sensing. Finally, with a refined training recipe (Section 3.3), we introduce the spatially-grounded Cambrian-*S* model family (Section 3.4).

## 3.1 BASE MODEL TRAINING: UPGRADED CAMBRIAN-1

We begin by developing an image-based MLLM base model, as robust semantic perception forms the foundation for higher-level spatial cognition. We follow the two-stage training pipeline of Cambrian-1 (Tong et al., 2024a). We upgrade the visual encoder to SigLIP2-SO400m (Tschannen et al., 2025) and the language model to the instruction-tuned Qwen2.5 (Yang et al., 2024a). For the vision-language connector, we adopt a simple two-layer MLP primarily for its computational efficiency. Full implementation details are provided in Appendix F.

## 3.2 SPATIAL VIDEO DATA CURATION: VSI-590K

It is well recognized that data quality and diversity play a critical role in the training of MLLMs (Tong et al., 2024a; McKinzie et al., 2024). We hypothesize that the performance gap on VSI-Bench (Yang et al., 2024d)

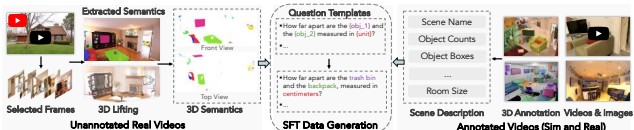

Figure 5: **VSI-590K data curation pipeline.**

comes mainly from the lack of high-quality, spatially grounded data in current instruction-tuning datasets (Zhang et al., 2025b; Cui et al., 2024). To fill this gap, we build **VSI-590K**, a large-scale instruction-tuning dataset designed to improve visual-spatial understanding. We define 12 question types to construct a comprehensive and diverse set of questions for instruction tuning, please refer to Appendix E.1 for details about the question type definition and taxonomy. Our dataset is constructed from a diverse span of data sources and types (*i.e.*, simulated and real), please refer to Appendix E.2 for details about the data curation, processing, and QA construction pipeline.

**VSI-590K data source ablation.** To evaluate the effectiveness of VSI-590K, we perform an ablation study by finetuning the improved Cambrian-1 MLLM described in Section 3.1 with part of the video instruction tuning samples from LLaVA-Video-178K (Zhang et al., 2025b). This model serves as the *baseline* in Table 7. The contribution of each data source is evaluated by fine-tuning the model on individual datasets as well as their combination. The VSI-590K Full Mix achieves the highest overall performance on video spatial reasoning tasks, outperforming both the baseline and all single-source counterparts. **All sources contribute positively** after fine-tuning, though their effectiveness varies.

> 🔖 Data effectiveness: annotated real videos > simulated data > pseudo-annotated images.

This indicates that videos are inherently more informative than static images for spatial reasoning, as training exclusively on video data yields superior performance on *both* video- and image-based spatial reasoning benchmarks. These findings support the intuition that the temporal continuity and multi-view diversity of videos are key to developing robust spatial representations.

## 3.3 POST-TRAINING RECIPE FOR SPATIAL SENSING

We further analyze and ablate our video instruction-tuning pipeline, focusing on the roles of the pretrained base video model and the instruction-tuning dataset mixture. As shown in Table 11, we begin with four *base models* that represent a progressive increase in video understanding capability: **A1** is trained only with image-text alignment on Cambrian-1 alignment data. The language model is identical to base QwenLM as it is frozen during training. **A2** is finetuned with image instruction tuning on top of A1, essentially our improved Cambrian-1. **A3** is initialized from A2 and finetuned on 429K video instruction tuning data. **A4** is initialized from A2 and finetuned on 3M video data.

Table 2: **Comparison of Cambrian-S with other leading MLLMs.** Cambrian-S outperforms both proprietary and open-source models across benchmarks and model sizes. Detailed evaluation setups are listed in Appendix G.

| Model | Base LM | Video | | | | | | | | | | Image | | |
| | | VSI-Bench | VSI-Bench(Debiased) | Tomato | HourVideo | VideoMME | EgoSchema | VideoMMMU | LongVBench | MVBench | Percept Test | MMVP | 3DSR | CV-Bench |
|---|---|---|---|---|---|---|---|---|---|---|---|---|---|---|
| *Proprietary Models* | | | | | | | | | | | | | | |
| GPT-4o | UNK. | 34.0 | - | 37.7 | 37.2 | 71.9 | - | 61.2 | 66.7 | - | - | 66.0 | 44.2 | - |
| Gemini-1.5-Pro | UNK. | 45.4 | 40.1 | 36.1 | 37.3 | 75.0 | 72.2 | 53.9 | 64.0 | - | - | 51.3 | - | - |
| Gemini-2.5 Pro | UNK. | 51.5 | 49.1 | - | - | - | - | 83.6 | 67.4 | - | - | 51.3 | - | - |
| *Open-Source Models* | | | | | | | | | | | | | | |
| LLaVA-Video-7B | Qwen2-7B | 35.6 | 30.7 | 22.5 | 28.6 | 63.3 | 57.3 | 36.1 | 58.2 | 58.6 | 67.9 | - | - | 75.7 |
| Qwen-VL-2.5-7B | Qwen2.5-7B | 33.5 | 29.6 | - | - | 65.1 | 65.0 | 47.4 | 56.0 | 69.6 | - | 56.7 | 48.4 | - |
| InternVL3.5-8B | Qwen3-8B | 56.3 | 49.7 | - | - | 66.0 | 61.2 | 49.0 | 62.1 | 72.1 | - | 56.0 | - | - |
| **Cambrian-S-7B** | Qwen2.5-7B | **67.5** | **59.9** | **27.0** | **36.5** | 63.4 | **76.8** | 38.6 | 59.4 | 64.5 | **69.9** | **60.0** | **54.8** | **76.9** |
| Qwen2.5-VL-3B | Qwen2.5-3B | 26.8 | 22.7 | - | - | 61.5 | - | - | 54.2 | - | 66.9 | 39.3 | - | - |
| **Cambrian-S-3B** | Qwen2.5-3B | **57.3** | **49.7** | **25.4** | **36.8** | 60.2 | **73.5** | 25.2 | 52.3 | **60.2** | 65.9 | **50.0** | **50.9** | **75.2** |
| SmolVLM2-2.2B | SmolLM2-1.7B | 27.0 | 22.3 | - | - | - | 34.1 | - | - | 48.7 | 51.1 | - | - | - |
| InternVL3.5-2B | Qwen3-1.7B | 51.5 | 46.1 | - | - | 58.4 | 50.8 | - | 57.4 | 65.9 | - | 44.0 | - | - |
| **Cambrian-S-1.5B** | Qwen2.5-1.5B | **54.8** | **47.5** | **22.5** | **31.4** | 55.6 | **68.8** | 24.9 | 50.0 | 58.1 | **63.2** | 42.7 | **51.9** | **69.6** |
| SmolVLM2-0.5B | SmolLM2-360M | 26.1 | 23.1 | - | - | - | 20.3 | - | - | 43.7 | 44.8 | - | - | - |
| InternVL3.5-1B | Qwen3-0.6B | 49.9 | 41.8 | - | - | 51.0 | 41.5 | 33.0 | 53.0 | 61.0 | - | 32.0 | - | - |
| **Cambrian-S-0.5B** | Qwen2.5-0.5B | **50.6** | **42.2** | **23.4** | **27.9** | 44.0 | **62.4** | 15.7 | 44.0 | 51.8 | **56.0** | 26.0 | **48.5** | **59.8** |

We then finetune these models using two different data recipes: (1) VSI-590K only, and (2) VSI-590K mixed with a similar amount of general video instruction tuning data.

> 🔖 Stronger base model with greater exposure to general video data leads to improved spatial sensing.

As shown in Table 11, SFT with a stronger base model, one that performs well on general video benchmarks such as VideoMME and EgoSchema, leads to enhanced spatial understanding. This highlights the importance of broad exposure to general video data during base model training.

> 🔖 Mixing general video data prevents the generalization loss caused by in-domain SFT.

Furthermore, while in-domain SFT solely on VSI-590K achieves the highest performance on VSI-Bench, it results in a noticeable decline on general video benchmarks. However, this performance drop can be effectively mitigated by training on a data mix that includes general videos.

## 3.4 CAMBRIAN-S: SPATIALLY-GROUNDED MLLMS

Building on all the previous insights, we develop **Cambrian-S**, a family of spatially-grounded models with varying LLM scales: 0.5B, 1.5B, 3B, and 7B parameters. These models are built

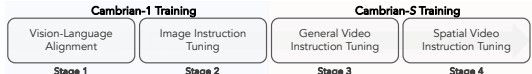

Figure 6: **Overall Cambrian-S training pipeline.**

through a four-stage training pipeline specifically designed to first establish general semantic perception and then develop specialized spatial sensing skills, as illustrated in Fig. 6. The first two stages adhere to the Cambrian-1 framework to develop strong image understanding capabilities. In stage 3, we extend the models to video by conducting general video instruction tuning on Cambrian-S-3M, a curated dataset composed of 3 million samples (see detailed composition in Fig. 15). In the final and crucial stage 4, the models are trained for spatial sensing. Here, we finetune the models on a blended corpus combining our specialized VSI-590K with a proportional subset of the general video data, following the setup described in Table 11. Complete training details are provided in Appendix F.3.

## 3.5 EMPIRICAL RESULTS: IMPROVED SPATIAL COGNITION

**Improved spatial cognition.** As shown in Table 2, our models achieve state-of-the-art performance in visual-spatial understanding in video. Cambrian-S-7B achieves 67.5% on VSI-Bench, significantly outperforming all open-source models and surpassing the proprietary Gemini-2.5-Pro by over 16 absolute points. Since our work in this section can be viewed as a data scaling effort, a natural question is: *are the performance improvements simply due to broader data coverage (including more diverse visual configurations and question–answer pairs), or has the model actually developed stronger spatial cognition?* First, we emphasize that there is no data overlap between VSI-590K and the benchmark datasets. Although some datasets originate from the same sources (*e.g.*, from ScanNet), we only use the training split, while the benchmarks use validation and test splits. Moreover, we observe clear signs of generalization in spatial reasoning. For example, in "Route Planning" subtask (absents from VSI-590K because of the high annotation cost), Cambrian-S-7B still performs strongly, showing pronounced scaling behavior with increasing model size too (see Table 12). Furthermore, our training approach proves highly effective even with smaller model sizes: our smallest *0.5B model* achieves performance comparable to Gemini-1.5 Pro on VSI-Bench. Importantly, this emphasis on

Table 3: **Cambrian-*S*-7B results on VSI-Super.** Despite strong performance on VSI-Bench, accuracy on VSR drops sharply from 38.3% (10 min) to 0.0% (>60 min), and VSC completely fails.

| | VSR | | | | | VSC | | | |
|---|---|---|---|---|---|---|---|---|---|
| Eval Setup | 10 min | 30 min | 60 min | 120 min | 240 min | 10 mins | 30 min | 60 min | 120 min |
| Uni. Sampling, 128F | 26.7 | 21.7 | 23.3 | 30.0 | 28.2 | 16.0 | 0.0 | 0.0 | 0.0 |
| FPS Sampling, 1FPS | 38.3 | 35.0 | 6.0 | 0.0 | 0.0 | 0.6 | 0.0 | 0.0 | 0.0 |

spatial reasoning does not come at the expense of general capabilities: Cambrian-*S* continues to deliver competitive results on standard video benchmarks (see Table 14 for complete results).

> 🔖 Cambrian-*S* achieves state-of-the-art spatial sensing with robust generalization to unseen spatial question types, while staying competitive in general video understanding.

**Robust spatial reasoning on VSI-Bench-Debiased.** To investigate whether Cambrian-*S* learns to reason visually, we evaluate it on VSI-Bench-Debiased (Brown et al., 2025), a benchmark specifically designed to eliminate language shortcuts through debiasing. As shown in Table 2, although performance decreases by about 8% compared to standard VSI-Bench, our models still outperform proprietary counterparts, demonstrating robust visual-spatial reasoning capabilities and confirming that our training extends beyond language-based learning.

**Results on VSI-Super: limitations in continual spatial sensing.** Despite its strong performance on spatial reasoning tasks in short, pre-segmented videos from VSI-Bench, Cambrian-*S* isn't well-equipped for continual spatial sensing. This limitation is evident in two ways. First, its performance deteriorates significantly on long videos. As shown in Table 3, when evaluated on VSI-Super with 1 FPS sampling in a streaming-style setup, scores drop steadily from 38.3% to 6.0% as video length increases from 10 to 60 minutes, and the model fails completely on videos longer than 60 minutes. Second, the model has difficulty generalizing to new test scenarios. Although trained on multi-room house tour videos, it fails to handle unseen examples with just a few additional rooms. This issue isn't simply about context length: performance drops even on short 10-minute videos that fit comfortably within model's context window. These results highlight that a purely data-driven approach, no matter how much data or engineering effort is invested, faces fundamental limits. Addressing these limitations calls for a paradigm shift toward AI systems that can actively model and anticipate the world while organizing their experiences more efficiently, which we explore next.

> 🔖 Scaling data and models is essential, but alone it cannot unlock true spatial supersensing.

## 4 PREDICTIVE SENSING AS A NEW PARADIGM

Performance of both Gemini-2.5-Flash (Table 1) and Cambrian-*S* (Table 3) reveals a fundamental paradigm gap: scaling data and context alone is insufficient for supersensing. We propose *predictive sensing* as a path forward, where models learn to anticipate their sensory input and construct internal world models to handle unbounded visual streams. We prototype this concept via a self-supervised next-latent-frame prediction approach (Section 4.1). The resulting prediction error serves as a control signal for two key capabilities: memory management to selectively retain important information (Section 4.2), and event segmentation to partition unbounded streams into meaningful chunks (Section 4.3). We demonstrate through two case studies on VSI-Super that this approach substantially outperforms strong long-context and streaming video model baselines.

### 4.1 PREDICTIVE SENSING VIA LATENT FRAME PREDICTION

We implement our predictive sensing paradigm through a lightweight, self-supervised module called the Latent Frame Prediction (LFP) head, which is trained jointly with the primary instruction-tuning objective. This is achieved by modifying the stage 4 training recipe as follows: **Latent frame prediction head:** We introduce an LFP Head, a two-layer MLP that operates in parallel with the language head, to predict the latent representation of the subsequent video frame. This architecture is illustrated in the top left of Fig. 7. **Learning objectives:** To optimize

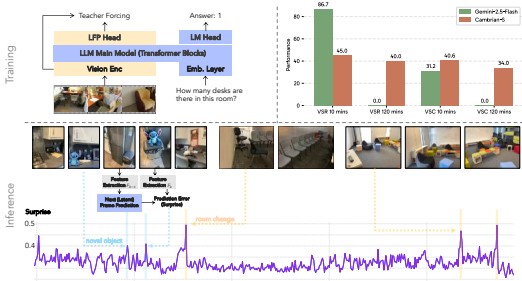

Figure 7: **Training and inference pipeline for the latent frame prediction (LFP) approach.**

the LFP head, we introduce two auxiliary losses, mean squared error (MSE) and cosine distance,

which measure the discrepancy between the predicted latent feature and the ground truth feature of the next frame. A weighting coefficient balances the LFP loss against the primary instruction-tuning next token prediction objective. **Data for LFP training:** We augment stage 4 data with a 290K video subset from VSI-590K used exclusively for the LFP objective. Unlike instruction tuning, these videos are sampled at 1 FPS to ensure uniform temporal spacing for latent frame prediction.

During this modified stage 4 finetuning, we train the connectors, language model, and both the language and LFP heads jointly in an end-to-end manner, while keeping the SigLIP vision encoder frozen. All other training settings remain unchanged. For brevity, we still denote the model jointly optimized with the LFP objective as Cambrian-*S* in subsequent experiments.

**Inference: Estimating surprise via prediction error.** During inference, we leverage the trained LFP head to evaluate the "surprise" (or, Violation-of-Expectation (Burgoon & Hale, 1988)) for every incoming visual sensory input. Specifically, during inference, video frames are fed into Cambrian-*S* at a constant sampling rate. Unless otherwise noted, the videos in the following experiments are sampled at 1 FPS before being input into the model. As the model receives incoming video frames, it continuously predicts the latent features of the next frame. We then measure the *cosine distance* between the model's prediction and the actual ground truth feature of that incoming frame. This distance serves as a quantitative measure of surprise: a larger value indicating a greater deviation from the model's learned expectations. This surprise score acts as a powerful, self-supervised guidance signal for the downstream tasks explored next.

## 4.2 CASE STUDY I: SURPRISE-DRIVEN MEMORY MANAGEMENT SYSTEM FOR VSR.

**Surprise-driven memory management system.** We build one memory management system that dynamically compresses and consolidates visual streams based on the estimate of "surprise". As shown in Fig. 8-(a), we encode incoming frames using sliding window attention with fixed window size. The latent frame prediction module then measures a "surprise level" and assigns it to each frame's KV caches. Frames with a surprise level below a predefined threshold undergo 2× compression before being pushed into long-term memory. To maintain a stable GPU memory footprint, this long-term memory is constrained to a fixed size by a consolidation function that, once again, operates based on *surprise*: dropping or merging frames according to their surprise scores (see Fig. 8(b)). Finally, upon receiving a user query, the system retrieves the top-$K$ most relevant frames from the long-term memory by calculating the cosine similarity between the query and the stored frame features (see Fig. 8-(c)). See Appendix H.2 for more design details. While prior works have explored memory system designs for long videos (Song et al., 2024; Zhang et al., 2025a), our focus is on exploring prediction errors (*i.e.*, surprise) as guiding signals.

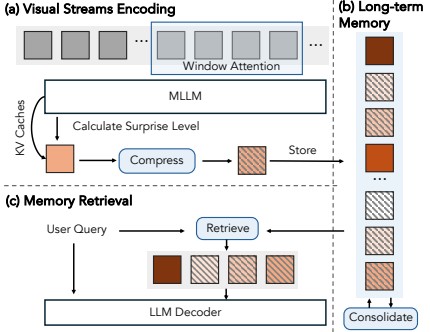

Figure 8: **Surprise-driven memory management framework for VSR**.

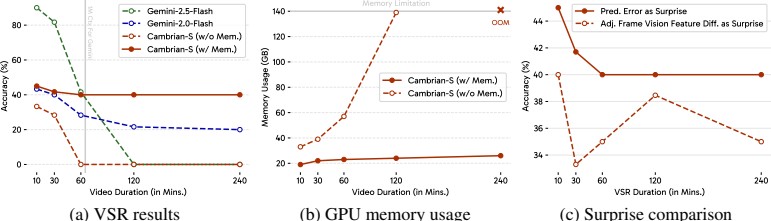

(a) VSR results  (b) GPU memory usage  (c) Surprise comparison

Figure 9: **Surprise-driven memory allows Cambrian-*S* to maintain strong performance (a) and stable GPU memory footprint (b).** (c) LFP prediction error is more robust than adjacent-frame difference as surprise.

**Results.** We compare Cambrian-*S* with and without the surprise-based memory system against two advanced proprietary models, Gemini-1.5-Flash (Team et al., 2024) and Gemini-2.5-Flash (Comanici et al., 2025), on the VSR benchmark. As shown in Fig. 9a, Cambrian-*S* (*w/* Mem.) outperforms both Gemini-1.5-Flash and Cambrian-*S* (*w/o.* Mem.) at all video lengths, demonstrating consistent spatial sensing performance across video durations. Although Gemini-2.5-Flash yields strong results for videos within an hour, it fails to process longer inputs. In addition to maintaining high accuracy, Cambrian-*S* (*w/* Mem.) also maintains stable GPU memory usage across different video lengths

(Fig. 9b). This demonstrates that surprise-based memory effectively compresses redundant data without losing critical information. We include two long-video baselines, MovieChat (Song et al., 2024) and Flash-VStream(Zhang et al., 2025a), for comparison in Table 17.

**Ablation on surprise measurement.** Here, we compare our design, prediction error as surprise, to another baseline: SigLIP2 feature difference as surprise. As shown in Fig. 9c, using prediction error consistently outperforms adjacent-frame similarity across different video durations.

> 🔖 Predictive sensing provides a more principled approach to modeling the spatiotemporal dynamics of video data than static similarity measures based on per-frame features.

### 4.3 CASE STUDY II: SURPRISE-DRIVEN CONTINUAL VIDEO SEGMENT FOR VSC.

**Surprise-driven event segmentation.** In the VSI-SUPER Count (VSC) benchmark, we examine a simple setting where surprise is used to segment continuous visual input,

identifying scene changes as natural breakpoints that divide the video stream into *spatially coherent* segments. This approach also parallels human problem-solving: when counting objects across a large area, people typically focus on one section at a time before combining the results. This behavior is also related to the "doorway effect" (Radvansky et al., 2011) (*i.e.*, passing through a doorway creates a natural memory boundary).

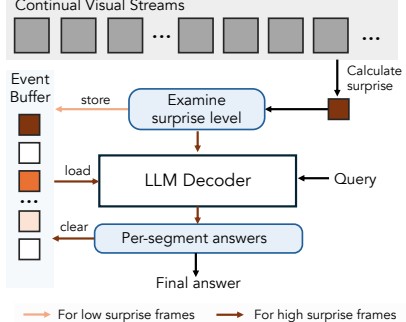

As illustrated in Fig. 10, the model continuously accumulates frame features in an event buffer. When a high-surprise frame is detected, the buffered features are summarized to produce a segment-level answer, and the buffer is cleared to start a new segment. This cycle repeats until

Figure 10: **Surprise-driven event segmentation framework for VSC.**

the end of the video, after which all segment answers are aggregated to form the final output.

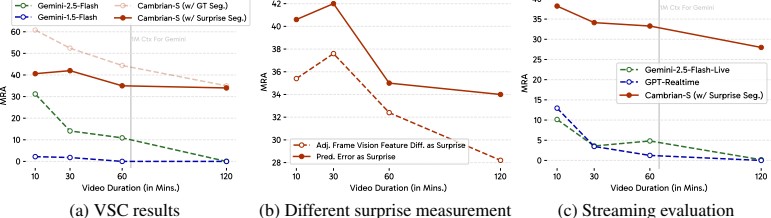

(a) VSC results      (b) Different surprise measurement      (c) Streaming evaluation

Figure 11: **Cambrian-*S* with surprise-driven event segmentation achieves consistently better results on both offline (a) and streaming setups (c).** (b) Pred. error as surprise outperforms adjacent-frame difference.

**Results.** Gemini-1.5-Flash attains near-zero performance on VSC (Fig. 11a), showing the task's difficulty. Although Gemini-2.5-Flash yields much better results on 10-minute videos, its performance declines rapidly on longer videos. In contrast, the surprise-driven event segmentation approach used by Cambrian-*S* (*w/ Surprise Seg.*) achieves higher and more stable performance across all video lengths. When the video is segmented using ground-truth scene transitions (*i.e.*, Cambrian-*S w/* GT Seg.), performance improves further, representing an approximate upper bound. A deeper analysis in Fig. 17 reveals that Gemini-2.5-Flash's predictions are confined to a limited range and do not scale as more objects appear in the video. In contrast, Cambrian-*S* (*w/ Surprise Seg.*) produces counts that exhibits a stronger correlation with the true object numbers, indicating better generalization.

**Ablation on surprise measurement.** We compare our surprise-driven approach to adjacent-frame feature similarity (Fig. 11b). For both methods, we report the best results after hyperparameter tuning.

**Evaluation in streaming setup.** As the correct answer in VSC evolves throughout the video, we create a streaming QA setup where the same question is asked at 10 different timestamps. The final performance is averaged across all queries. We benchmark against commercial MLLMs marketed for live visual input. As shown in Fig. 11c, although Gemini-Live and GPT-Realtime are intended for streaming scenarios, they achieve under 15% MRA on 10-minute videos and their performance declines to near zero on 120-minute streams. Cambrian-*S*, however, shows stronger performance, reaching 38% MRA on 10-minute streams and maintaining around 28% at 120 minutes.

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

## ACKNOWLEDGMENTS

We are grateful to Cambrian-1 (Tong et al., 2024a) for the excellent codebase, which served as the launching point for our research. Thanks to the TorchXLA team for helpful discussions on TPU, TorchXLA, and JAX distributed training infrastructure. We also thank Anjali Gupta, Sihyun Yu, Oscar Michel, Boyang Zheng, Xichen Pan, Weiyang Jin, and Arijit Ray for reviewing this manuscript and providing constructive feedback. This work was primarily supported by the Google TPU Research Cloud (TRC) program and the Google Cloud Research Credits program (GCP19980904). E.B. is supported by the DoD NDSEG Fellowship Program. S.X. acknowledges support from the MSIT IITP grant (RS-2024-00457882) and the NSF award IIS-2443404.

## APPENDIX

This appendix provides comprehensive implementation details, experimental results, and supplementary analyses supporting the main paper:

- § A provides a summarization and limitations of our current work and outlines potential directions for future explorations.
- § B provides a summarization of related work, including recent advancements in video MLLMs, streaming video understanding, visual spatial intelligence, and predictive modeling.
- § C presents detailed diagnostic test results for video MLLM benchmarks under different evaluation setups.
- § D describes the VSI-SUPER benchmark, including implementation details, visualizations, and streaming setups for both Recall and Count tasks.
- § E provides comprehensive documentation of the VSI-590K dataset, including question type taxonomy, QA-pair construction pipeline, ablation studies, and qualitative examples.
- § F details the Cambrian-*S* model architecture, training data mixture, training recipe across all four stages, and infrastructure setup.
- § G presents additional experimental results including detailed evaluation setups, performance on image and video benchmarks across all model scales, ablations on image-video data contributions, and analysis of the trade-off between spatial sensing and general video understanding.
- § H describes predictive sensing components, including latent frame prediction implementation details, memory framework design for VSI-SUPER Recall, agentic framework design for VSI-SUPER Count, and comparisons with existing long-video methods.

## A    DISCUSSION

In this paper, we highlight the importance of and propose a hierarchy for spatial *supersensing* capabilities in videos, arguing that achieving superintelligence requires AI systems to move beyond text-based knowledge and semantic perception, the current focus of most MLLMs, to also develop spatial cognition and predictive world models. To measure progress, we introduce VSI-SUPER and find that current MLLMs struggle with it. To test whether current progress is limited by data, we curate VSI-590K and train our spatially grounded MLLM, Cambrian-*S*, on it. Although Cambrian-*S* performs well on standard benchmarks, its results on VSI-SUPER reveal the limitations of the current MLLM paradigm. We prototype predictive sensing, using latent frame prediction and surprise estimation to handle unbounded visual streams. It improves Cambrian-*S* performance on VSI-SUPER and marks an early step toward spatial supersensing.

**Limitations and Future Work.** Our goal is to present a conceptual framework that encourages the community to reconsider the importance of developing spatial supersensing. As a long-term research direction, our current benchmark, dataset, and model design remain limited in quality, scale, and generalizability, and the prototype serves only as a proof of concept. Future work should explore more diverse and embodied scenarios and build stronger connections with recent advances in vision, language, and world modeling.

## B  RELATED WORK

**Video Multimodal Large Language Models**   The strong linguistic understanding capabilities of pretrained LLMs (Brown et al., 2020; Touvron et al., 2023a; Bai et al., 2023a; Touvron et al., 2023b), combined with the representational power of vision foundation models used as feature extractors (Radford et al., 2021; Zhai et al., 2023; Tschannen et al., 2025; He et al., 2022; Fan et al., 2025), have driven significant advances in extending these models beyond text to achieve semantic perception of visual content, primarily in the image domain (Hurst et al., 2024; Liu et al., 2023; Li et al., 2025a; Bai et al., 2023b; Tong et al., 2024a; Team et al., 2023; Chen et al., 2024d; Wang et al., 2024b; Li et al., 2023a). This momentum has spurred growing research into video-based MLLMs (Li et al., 2024f; 2025a; Zhang et al., 2025b; Song et al., 2024; Bai et al., 2025a; Zhu et al., 2025b; Zhang et al., 2023; Li et al., 2023b; Zohar et al., 2025; Marafioti et al., 2025), which are seen as a key step toward connecting multimodal intelligence with real-world applications such as embodied agents (Kim et al., 2024; Yang et al., 2024c). As emphasized throughout this paper, developing a truly capable supersensing system requires rethinking several core aspects, including how progress is benchmarked, what constitutes the right data, which architectural designs are most effective, and what modeling objectives best align with the system's goals.

**Streaming Video Understanding**   Video is a continuous and potentially infinite stream of visual signals. While humans process it effortlessly, its unbounded nature challenges video MLLMs because token lengths increase with duration, causing rising computational and storage costs. Recent work has explored several approaches to address this problem: *Efficient architectural design*. The quadratic cost of self-attention makes it hard to handle long videos. Recent methods (Li et al., 2024c; Ren et al., 2025) use simpler, faster architectures (Wang et al., 2020; Gu & Dao, 2024; Katharopoulos et al., 2020) that reduce computation and work better with longer inputs. *Context window expansion*. The fixed context length in pre-trained LLMs limits their understanding of long-term content. Recent work (Chen et al., 2025c; Zhang et al., 2024; Chen et al., 2025b) extends this window by careful system design, enabling models to handle and reason over longer video sequences. *Retrieval-augmented video understanding*. To process long videos, some approaches retrieve only the most relevant segments from a larger collection (Korbar et al., 2024; Pan et al., 2025; Wang et al., 2024c) and use them as context for further analysis.*Visual token reduction or compression*. Other methods shorten the input by reducing visual tokens across or within frames (Shen et al., 2025; Li et al., 2024e; Jiang et al., 2025; Li et al., 2025b; Chai et al., 2025), making it easier to handle long video sequences. While these methods improve performance, they largely treat continuous videos as standard sequence modeling problems, similar to text. We believe future MLLMs should build internal predictive models to efficiently process continuous visual streams, as humans do.

**Visual Spatial Intelligence**   Understanding spatial relationships from visual inputs is crucial for perceiving and interacting with the physical world. As multimodal models become more physically grounded, interest in spatial intelligence has surged, leading to new benchmarks (Yang et al., 2024d; Ramakrishnan et al., 2025; Yin et al., 2025; Majumdar et al., 2024; Yeh et al., 2025; Li et al., 2025c; Xu et al., 2025; Team et al., 2025) and research focused on enhancing models' spatial reasoning capabilities (Yang et al., 2025b; Ma et al., 2025; Ouyang et al., 2025; Du et al., 2024; Chen et al., 2024a; Cheng et al., 2024; Cai et al., 2025; Liu et al., 2025; Li et al., 2024b; Zhu et al., 2025a; Ray et al., 2025). In this paper, we study visual spatial intelligence through the concept of spatial supersensing in videos and explore ways to strengthen MLLMs' spatial reasoning by refining data curation, optimizing training strategies, and introducing new paradigms.

**Predictive Modeling**   A learned internal predictive model (Craik, 1967; Ha & Schmidhuber, 2018) allows an intelligent agent to represent and simulate aspects of its environment, enabling more effective planning and decision-making. Model predictive control (MPC) (Garcia et al., 1989) applies similar principles in control theory, leveraging internal forward models to anticipate future trajectories and select optimal actions in real time. This concept draws inspiration from how humans form mental models of the world (Rao & Ballard, 1999; Hohwy, 2013; Friston, 2010) and how these internal representations influence behavior (*e.g.*, *unconscious inference* (Von Helmholtz, 1867)), serving as simplified abstractions of reality that enable prediction and efficient action. A growing body of work has explored the idea of predictive modeling through self-supervised representation learning (Assran et al., 2023; 2025), and text- or action-conditioned video generation (Zhou et al.,

2025; Yang et al., 2024e; Bar et al., 2025; Chen et al., 2024b; Bai et al., 2025b; Garrido et al., 2025). In this paper, motivated by how humans leverage internal world models to process unbounded sensory input efficiently and effectively, we explore how to equip MLLMs with a similar predictive sensing capability.

## C  BENCHMARK DIAGNOSTIC TEST RESULTS

### C.1  DETAILED RESULTS OF CAMBRIAN-1-7B ON VIDEO BENCHMARKS

Table 4: **Detailed results of our improved Cambrian-1-7B on video MLLM benchmarks under different evaluation setups.**

| Evaluation Setups | VideoMME | EgoSchema | VideoMMMU | LongVideoBench | Tomato | MVBench | Perception Test | HourVideo | VSI-Bench | VSI-SUPER Recall | VSI-SUPER Count |
|---|---|---|---|---|---|---|---|---|---|---|---|
| *Chance-Level* | 25.0 | 20.0 | 14.0 | 25.0 | 22.0 | 27.3 | 33.3 | 20.0 | 34.0 | 25.0 | 0.0 |
| *Cambrian-1-7B (Our upgraded)* | | | | | | | | | | | |
| Blind Test | 31.2 | 31.9 | 25.0 | 42.5 | 7.8 | 19.6 | 40.7 | 24.3 | 17.4 | 20.0 | 0.0 |
| Single Frame | 41.6 | 44.0 | 29.0 | 46.9 | 15.8 | 46.1 | 52.1 | 27.7 | 20.4 | 19.7 | 0.0 |
| Multiple (32) Frames | 53.7 | 48.1 | 31.9 | 51.4 | 18.9 | 55.4 | 55.6 | 31.6 | 25.8 | 22.7 | 0.0 |
| (32) Frame Captions | 55.3 | 52.4 | 40.1 | 52.2 | 16.8 | 47.7 | 55.6 | 29.5 | 21.8 | 9.6 | 0.5 |

### C.2  BENCHMARK EXAMPLES

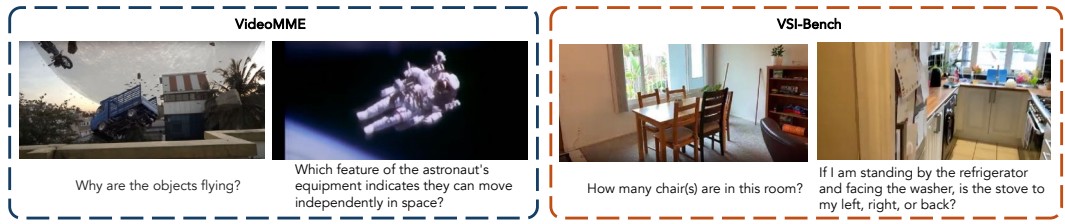

Figure 12: **Illustrations of how spatial sensing is conceptualized in current video benchmarks**. The left panel features examples from the "spatial reasoning" subcategory of VideoMME (Fu et al., 2025), including a question regarding gravity from Shutter Authority's "`What if the Moon Crashed into the Earth?`" and a question regarding astronaut gear from NASA's "`Astronaut Bruce McCandless II Floats Free in Space.`" In contrast, the right panel shows samples from VSI-Bench (Yang et al., 2024d), which highlight visual-spatial reasoning tasks such as object counting, identifying relative directions, route planning, and more.

## D  VSI-SUPER BENCHMARK

### D.1  VSI-SUPER RECALL

**Implementation details.**    To construct this benchmark, we begin with videos from the VSI-Bench collection (Yang et al., 2024d). Annotators select videos and manually insert an unusual object from a curated pool into four distinct frames using Gemini-2.0-Flash, focusing on placing the objects in plausible locations. For each insertion, the annotators record the object's location and its order of appearance. We then combine these edited clips with randomly sampled unedited videos to produce final videos with lengths of 10, 30, 60, 120, and 240 minutes. For each duration, we create 60 videos, each with one corresponding question. We downsample videos to 1 frame per second to ensure the model can always see the edited frames during inference.

**Visualization.**  We present qualitative examples of edited frames of our VSR video dataset in Fig. 18. The inserted objects appear visually plausible at their locations, which is a direct result of our high-quality annotations.

## D.2 VSI-SUPER COUNT

**Implementation Details.**  To build VSI-SUPER Count, we concatenate videos from VSI-Bench (Yang et al., 2024d) and sum their object counts to create a new ground truth. This process requires two additional normalization steps. First, we unify the object category labels from the different source datasets (*i.e.*, ScanNet (Dai et al., 2017), ScanNet++ (Yeshwanth et al., 2023), and ARKitScenes (Baruch et al., 2021)). Second, we address a data bias towards small object quantities by rebalancing the question-answer pairs to create a more uniform distribution of counts. The final benchmark includes videos with lengths of 10, 30, 60, and 120 minutes, each accompanied by 50 corresponding questions. Different from VSR, all videos in VSC are downsampled to 24 FPS.

**Streaming setups.**  For the streaming setup, we repeatedly query the total number of objects in a video at 10 distinct timestamps. To construct the ground truth at these query timestamps, we need to determine the first appearance time of each unique object in the video. To find these appearance times, we use the method proposed by the VSI-Bench (Yang et al., 2024d). This allows for the direct calculation of the ground truth object count at any given timestamp.

## D.3 VISUALIZATION OF GEMINI-2.5-FLASH'S PREDICTIONS ON VSC.

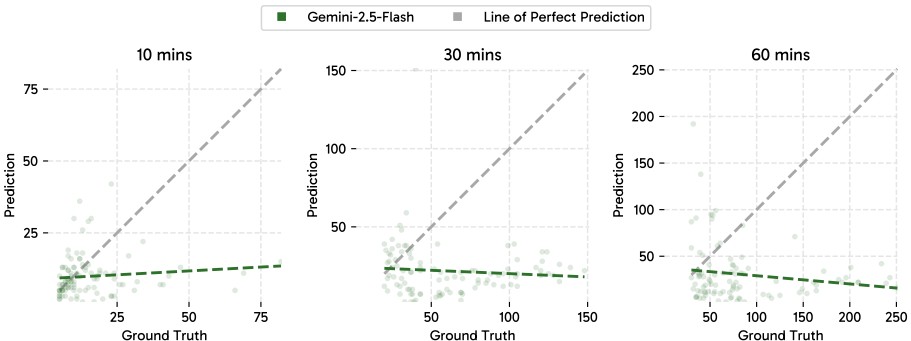

Figure 13: **Visualization of Gemini-2.5-Flash's predictions *v.s.* ground truth on VSC.** The model's predicted object counts saturate at small constant values and fail to scale with video length or true object counts, indicating limited generalization in counting and reliance on training distribution priors.

## E  VSI-590K DATASET

In this section, we provide more details for our VSI-590K dataset, including the question type definition, question-answer pair construction pipeline, and some examples for each data source.

### E.1  DETAILS OF QUESTION TYPE DEFINITION

**Taxonomy.**  When curating visual-spatial intelligence supervised fine-tuning datasets, an important perspective is how to define the question type. Inspired by VSI-Bench (Yang et al., 2024d), we expand its task definition in a more systematic manner. As shown in Table 5, we distinguish these question types in four perspectives:

- `Spatial-temporal attributes`: We categorize questions into five distinct spatial-temporal attribute types: size (comparing or measuring object/space dimensions), direction (orientation in space), count (enumeration of objects), distance (proximity between objects), and appearance order (temporal sequence of objects appearing in videos).

- `Relative versus absolute:` Questions are classified as relative when they involve comparison between multiple objects (*e.g.*, "which is larger?"), or absolute when they require specific measurements or quantities (*e.g.*, "what is the height in meters?"). This distinction applies across most attribute types.
- `Perspective taking:` This dimension captures the viewpoint from which spatial relationships are evaluated. Questions may be posed from the camera's perspective (*e.g.*, "from the camera's perspective, is the object on the left or right?") or from the perspective of specific objects in the scene (*e.g.*, "facing the object$_1$ from object$_2$...")
- `Modality:` Questions are categorized based on whether they can be answered using static images only, or require dynamic video information. Some attribute types, like appearance order, are only applicable to videos, while others like size can be questioned in either modality.

Additionally, following VSI-Bench, we also categorize our question types into three different groups (*i.e.*, *Configuration*, *Measurement*, or *Spatiotemporal*) according to their different spatiotemporal characteristics.

Table 5: **Taxonomy of spatiotemporal question types in VSI-590K.** Questions are stratified along five axes: attribute type, relative vs. absolute (Rel./Abs.), perspective, modality (V: video, I: image), and group. An example question template is provided for each type.

| Types | Rel./Abs. | Perspective | Modality | Group | Example template |
|---|---|---|---|---|---|
| Size | Rel. | — | V & I | Configuration | "Between {object$_1$} and {object$_2$}, which is larger?" |
| | Abs. | — | V & I | Measurement | "What is the height of the {object} in {unit}?" |
| | Abs. | — | V & I | Measurement | "What is the room's size in {unit}?" |
| Direction | Rel. | Camera | I | Configuration | "From the camera's perspective, is the {object} on the left or the right?" |
| | Rel. | Object | V & I | Configuration | "Facing the {object$_1$} from the {object$_2$}, would the {object$_3$} be placed left, right, or back?" |
| | Abs. | Object | V & I | Measurement | "Standing at {object$_1$}, facing toward {object$_2$}, how far clockwise do I rotate (in degrees) to see the {object$_3$}?" |
| Count | Rel. | — | V & I | Configuration | "Are there fewer {object$_1$} than {object$_2$} ?" |
| | Abs. | — | V & I | Measurement | "How many {object} are present?" |
| Distance | Rel. | Camera | I | Configuration | "Which object is closer to the camera, the {object_1} or the {object_2}?" |
| | Rel. | Object | V & I | Configuration | "Which is nearer to the {object_3}, the {object_1} or the {object_2}?" |
| | Abs. | Object | V & I | Measurement | "What is the distance between the {object_1} and the {object_2} in {unit}?" |
| Appr. Order | — | — | V | Spatiotemporal | "Determine how {object_1}, {object_2}, {object_3}, and {object_4} are ordered by their initial appearances in the video" |

### E.2 DETAILED QA-PAIR CONSTRUCTION PIPELINE

We introduce the concrete pipeline used for curating VSI-590K here.

**3D-annotated real videos.** For the 3D-annotated real videos, we follow the practice established by Thinking in Space (Yang et al., 2024d). We begin by researching all publicly available datasets containing both 3D instance-level annotations and video or panorama images. From these datasets, we extract key information including *object counts*, *object bounding boxes*, and *room size* measurements, which we then standardize into a unified format. Afterward, this structured information is incorporated into augmented question templates to create paired question-answer sets.

**3D-annotated simulated videos and images.** For simulated data, which inherently contains rich annotations, we followed a procedure similar to that used for 3D-annotated real videos. As for ProcTHOR (Deitke et al., 2022), our primary effort is generating 3D scenes with randomly placed agents to render traverse videos. For Hypersim (Roberts et al., 2021), which provides image-level rather than scene-level 3D annotations, we utilize individual images with their corresponding 3D annotations. In both cases, we extract the necessary information, convert it to our designed unified format, and incorporate it into augmented question templates, following the same approach used for 3D-annotated real videos.

**Unannotated web-crawled real videos.** For unannotated web-crawled real videos, as shown in Algorithm 1, we implement a multi-stage processing pipeline. We begin by sampling frames at regular intervals and filtering out blurry images. For each valid frame, we employ the open-vocabulary object detector Grounding-DINO (Liu et al., 2024c) with predefined categories of interest. When a frame contains sufficient valid objects, we use SAM2 (Ravi et al., 2025) to extract instance-wise semantic masks. Besides, to transform 2D image content into 3D representations, we employ VGGT (Wang et al., 2025) to extract 3D point sets for each image and integrate them with the previously generated instance masks. Notably, we apply an erosion algorithm to refine the instance masks, which mitigates inaccurate point cloud estimations at object boundaries. This pipeline has enabled us to create pseudo-annotations from approximately 19,000 room tour videos from YouTube and robotic learning datasets, yielding diverse spatial question-answer pairs across various room types and layouts without manual 3D annotations. By processing individual frames rather than complete videos, our pipeline ensures higher quality semantic extraction and more reliable reconstruction results, avoiding the noise and inconsistent issues typically encountered when applying reconstruction and semantic extraction techniques to entire video sequences.

### E.3 DATA STATISTICS

We provide detailed data statistics for VSI-590K in Table 6 and Fig. 14.

Table 6: **Data statistics for VSI-590K.** We collect data from 10 sources with different video types and annotations to improve diversity.

| Dataset | # Videos | # Images | # QA Pairs |
|---|---|---|---|
| *Annotated Real Videos* | | | |
| S3DIS (Armeni et al., 2016) | 199 | - | 5,187 |
| Aria Digital Twin (Pan et al., 2023) | 183 | - | 60,207 |
| ScanNet (Dai et al., 2017) | 1,201 | - | 92,145 |
| ScanNet++ V2 (Yeshwanth et al., 2023) | 856 | - | 138,701 |
| ARKitScenes (Baruch et al., 2021) | 2,899 | - | 57,816 |
| *Simulated Data* | | | |
| ProcTHOR (Deitke et al., 2022) | 625 | - | 20,092 |
| Hypersim (Roberts et al., 2021) | - | 5,113 | 176,774 |
| *Unannotated Real Videos* | | | |
| YouTube Room Tour | - | 20,100 | 20,100 |
| Open X-Embodiment (O'Neill et al., 2024) | - | 14,801 | 14,801 |
| AgiBot-World (Bu et al., 2025) | - | 4,844 | 4,844 |
| **Total** | **5,963** | **44,858** | **590,667** |

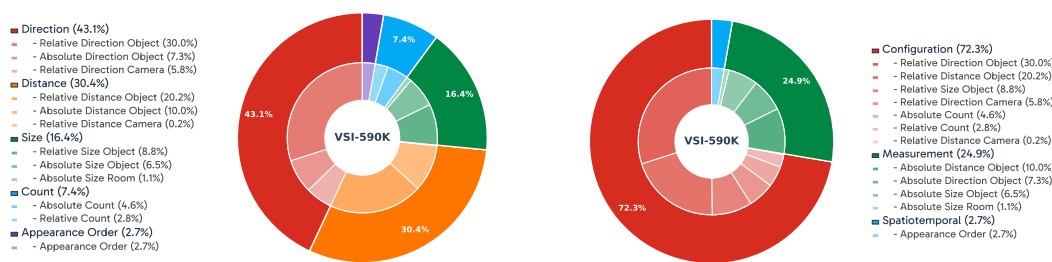

Figure 14: **VSI-590K statistics.** QAs are grouped by: question types (left) and task groups (right).

### E.4 ADDITIONAL ABLATION STUDY

Table 7 ablates the contributions of different data sources in VSI-590K. Table 8 presents an ablation study on how different task groups affect the model's spatial sensing capability. Our results show that all three task groups—configuration, measurement, and spatiotemporal—are integral, as removing any one of them degrades performance. We further assess spatial reasoning using the held-out *Route*

Table 7: **Contributions of Different Data Sources in the VSI-590K Mixture.** This table illustrates the impact of different data sources on VSI-Bench performance. The combined dataset, VSI-590K Full Mix, achieves the best overall results. Among individual sources, annotated real video datasets contribute the most significant improvements, followed by simulated videos, and then pseudo-annotated images.

| VSI Data Mixture | Image | | | VSI-Bench (Video) | | | | | | | | |
|---|---|---|---|---|---|---|---|---|---|---|---|---|
| | MMVP | 3DSR | CV-B | Avg | Obj Ct | Abs Dst | Obj Sz | Rm Sz | Rel Dst | Rel Dir | Rte Pln | Ap Ord |
| Baseline | 52.7 | 54.5 | 73.5 | 28.5 | 18.1 | 20.0 | 36.0 | 22.2 | 42.9 | 31.3 | 24.6 | 33.0 |
| *Real Videos* | | | | | | | | | | | | |
| + S3DIS | 54.0 | 54.9 | 75.3 | 41.6 | 63.8 | 21.0 | 44.9 | 37.0 | 43.8 | 47.4 | 34.0 | 41.1 |
| + ADT | 50.6 | 56.5 | 77.5 | 41.0 | 51.0 | 29.8 | 52.5 | 40.2 | 42.3 | 38.8 | 34.0 | 39.8 |
| + ARKitScenes | 50.0 | 56.7 | 77.3 | 51.0 | 70.2 | 32.7 | 64.5 | 60.0 | 55.1 | 45.2 | **37.1** | 43.5 |
| + ScanNet | 54.7 | **57.7** | 77.5 | 56.3 | 70.9 | 37.9 | 67.5 | 59.3 | 57.0 | 46.7 | 35.1 | 76.1 |
| + ScanNet++ V2 | 52.7 | 57.3 | 77.5 | 56.3 | 72.5 | 40.7 | 65.7 | 56.9 | 59.7 | 47.1 | 31.4 | 76.2 |
| *Simulated Videos* | | | | | | | | | | | | |
| + ProcThor | 53.3 | 55.7 | 74.9 | 36.4 | 21.0 | 29.7 | 49.3 | 3.8 | 52.3 | 45.7 | 30.4 | 58.7 |
| + HyperSim | 52.0 | 56.0 | **79.7** | 45.6 | 67.8 | 32.0 | 59.3 | 36.4 | 53.2 | 47.0 | 32.5 | 36.6 |
| *Pseudo-Annotated Images* | | | | | | | | | | | | |
| + YTB RoomTour | 55.3 | 52.6 | 75.0 | 32.5 | 43.4 | 25.8 | 24.2 | 27.3 | 38.7 | 31.4 | 28.4 | 40.9 |
| + OXE & AGIBot | **56.0** | 54.4 | 72.5 | 30.6 | 40.3 | 23.1 | 27.9 | 26.6 | 38.0 | 22.8 | 32.0 | 33.8 |
| **Full Mix** | 54.7 | 54.0 | 77.9 | **63.2** | **73.5** | **49.4** | **71.4** | **70.1** | **66.9** | **61.5** | 36.6 | **76.6** |

Table 8: **Ablation study on VSI-590K task groups.** We study models' performance change when one certain task group are omitted from the training data.

| VSI-590K Mixture | VSI-Bench | | | | | | | | |
|---|---|---|---|---|---|---|---|---|---|
| | Avg | Obj Ct | Abs Dst | Obj Sz | Rm Sz | Rel Dst | Rel Dir | Rte Pln | Ap Ord |
| All | 63.2 | 73.5 | 49.4 | 71.4 | 70.1 | 66.9 | 61.5 | 36.6 | 76.4 |
| *w/o.* Configuration | 51.9 | 46.2 | 43.0 | 70.4 | 66.0 | 48.0 | 36.8 | 27.3 | 77.3 |
| *w/o.* Measurement | 49.7 | 74.5 | 19.1 | 31.1 | 38.5 | 63.9 | 55.6 | 35.1 | 79.5 |
| *w/o.* Spatiotemporal | 58.1 | 73.7 | 47.7 | 70.9 | 65.2 | 68.3 | 58.9 | 32.5 | 47.6 |

*Plan* subtask and find that the configuration group is the most influential, whereas the measurement group is the least. We attribute this outcome to the fact that route planning requires a holistic understanding of the spatial layout, which is more explicitly provided by configuration QA pairs compared to measurement and spatiotemporal tasks.

### E.5 EXAMPLES OF VSI-590K

To better illustrate VSI-590K, we provide qualitative visualization results in Figs. 19 to 25. These visualizations demonstrate that VSI-590K delivers great diversity and quality for spatial question-answering supervised fine-tuning.

## F CAMBRIAN-*S* IMPLEMENTATION DETAILS

In this section, we provide holistic training details of our Cambrian-*S* models.

### F.1 MODEL ARCHITECTURE

Following the original Cambrian-1 (Tong et al., 2024a) and common practices in most MLLMs (Liu et al., 2023; Li et al., 2025a), our model (both our upgraded Cambrian-1 and Cambrian-*S*) integrates a pre-trained vision encoder, a pre-trained language model as the decoder, and a vision-language connector to bridge these two modalities. Specifically, we employ SigLIP2-So400M (Tschannen et al., 2025) as the vision encoder. This encoder was trained using a combination of losses: text next-token-prediction (LocCa (Wan et al., 2024)), image-text contrastive (or sigmoid (Radford et al., 2021; Zhai et al., 2023)), and masked self-prediction (SILC (Naeem et al., 2024)/TIPS (Maninis

---

**Algorithm 1: QA generation pipeline for unannotated web-scrawled videos**

---

**Input:** Video sequence $V$, valid category list $\mathcal{C}_{\text{valid}}$, invalid category list $\mathcal{C}_{\text{invalid}}$, sampling interval $\Delta t$, blur threshold $\tau_{\text{blur}}$, minimum object count $\theta_{\text{min}}$, minimum 3D point count $\theta_{\text{3D}}$, erosion kernel $K_{\text{erosion}}$

**Output:** Selected frame set $\mathcal{F}$, Question-answer pairs $\mathcal{Q}$

1  Initialize $\mathcal{F} \leftarrow \emptyset$, $\mathcal{Q} \leftarrow \emptyset$;
2  $\mathcal{S} \leftarrow \text{SampleFrames}(V, \Delta t)$ ;                          // Sample frames at interval $\Delta t$
3  **foreach** *frame* $f \in \mathcal{S}$ **do**
4     **if** *BlurDetection*$(f) > \tau_{blur}$ **then**
5        $\lfloor$ **continue**;
6     $\mathcal{O} \leftarrow \text{GroundingDINO}(f, \mathcal{C}_{\text{valid}} \cup \mathcal{C}_{\text{invalid}})$ ;      // Detect objects from both category lists
7     **if** $\exists o \in \mathcal{O} : category(o) \in \mathcal{C}_{invalid}$ **then**
8        $\lfloor$ **continue**;
9     $\mathcal{O}_{\text{valid}} \leftarrow \{o \in \mathcal{O} : \text{category}(o) \in \mathcal{C}_{\text{valid}}\}$;
10     **if** $|\mathcal{O}_{valid}| < \theta_{min}$ **then**
11        $\lfloor$ **continue**;
12     $\mathcal{M} \leftarrow \emptyset$ ;                                      // Initialize mask set
13     **foreach** *object* $o \in \mathcal{O}_{valid}$ **do**
14        $b \leftarrow \text{GetBoundingBox}(o)$;
15        $m \leftarrow \text{SAM2}(f, b)$ ;                          // Generate mask using SAM2
16        $m' \leftarrow \text{Erode}(m, K_{\text{erosion}})$ ;                  // Apply erosion on the masks
17        $\lfloor$ $\mathcal{M} \leftarrow \mathcal{M} \cup \{m'\}$;
18     $\mathcal{P}_{\text{map}} \leftarrow \text{VGGT}(f)$ ;                        // Generate 3D point map using VGGT
19     $\mathcal{P} \leftarrow \emptyset$ ;                                      // Initialize 3D point set
20     **foreach** *mask* $m \in \mathcal{M}$ **do**
21        $P \leftarrow \text{ExtractMaskedPoints}(m, \mathcal{P}_{\text{map}})$ ;    // Extract 3D points covered by mask
22        **if** $|P_{valid}| \geq \theta_{3D}$ **then**
23           $\lfloor$ $\mathcal{P} \leftarrow \mathcal{P} \cup \{P\}$;
24     **if** $|\mathcal{P}| > 0$ **then**
25        $q \leftarrow \text{QAGenerator}(\mathcal{P})$ ;                    // Generate QA pairs from 3D geometry
26        $\mathcal{Q} \leftarrow \mathcal{Q} \cup \{q\}$;
27        $\mathcal{F} \leftarrow \mathcal{F} \cup \{f\}$;
28  **Return** $\mathcal{F}$, $\mathcal{Q}$;

---

et al., 2024)). For the language model, we utilize the instruction-tuned Qwen2.5 LLMs (Yang et al., 2024a). Unlike Cambrian-1, which used SVA for a deeper vision-language fusion, we employ a simpler GELU-activated (Dauphin et al., 2017) two-layer MLP as the vision-language connector to maintain a balance between performance and efficiency.

## F.2  TRAINING DATA MIXTURE

As mentioned in Section 3.4, our Cambrian-*S* models are trained with four training stages (See Fig. 6). For the first two stages (*i.e.*, vision-language alignment stage and image instruction tuning stage), we refer readers to Cambrian-1 (Tong et al., 2024a) for the detailed training data mixture. In the third stage, we finetune the image instruction-tuned models Cambrian-*S*-3M, and during the last stage, we conduct spatial video instruction tuning by finetuning the model on VSI-590K. Cambrian-*S*-3M is our curated video instruction tuning dataset with around 3M video QA samples, built upon a set of open-sourced video datasets (*e.g.*, LLaVA-Video (Zhang et al., 2025b), ShareGPT4o (Cui et al., 2024),VideoChat2 (Li et al., 2024d), MovieChat (Song et al., 2024), EgoIT (Yang et al., 2025a), Perception Test (Patraucean et al., 2023), Vript (Yang et al., 2024b),VideoChatGPT-Plus (Maaz et al., 2024), Ego4D (Grauman et al., 2022), HowTo100M (Miech et al., 2019),HD-VILA (Xue et al., 2022), HTStep (Afouras et al., 2023), TimeIT (Ren et al., 2024), HowToInterlink7M (Wang et al., 2024a), GUI-World (Chen et al., 2025a), Video-Localized-Narratives (Voigtlaender et al., 2023), and *etc.*). We detail its composition in Fig. 15.

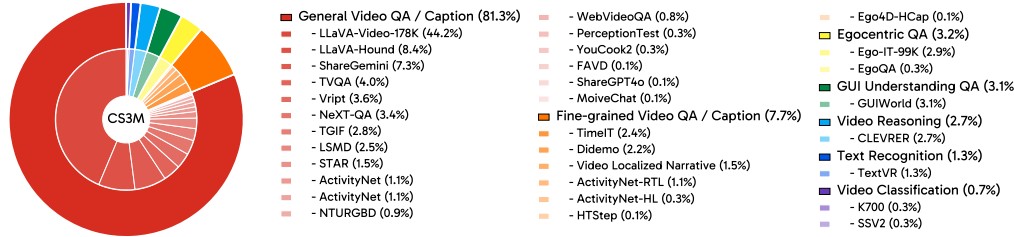

Figure 15: **General video instruction tuning datasets of Cambrian-*S*-3M, used in Cambrian-*S* stage 3 & 4 training.**

Table 9: **Training configuration for stage 1 and stage 2.**

|  | Stage 1 (Vision-Language Alignment) | Stage 2 (Image Instruction Tuning) |
|---|---|---|
| *Model* |  |  |
| Vision Encoder | SigLIP2-So400M | |
| Language Decoder | Qwen2.5-0.5B, 1.5B, 3B, 7B-Instruct | |
| VL-Connector | 2×MLP-GELU | |
| *Data Recipe* |  |  |
| Data | Cambrian-Alignment-2.5M | Cambrian-7M |
| Image Resolution | Pad (384×384) | AnyRes (Up to 9 sub-images) |
| # of Tokens per Image | 729 | Up to 7,290 |
| *Training Recipe* |  |  |
| Max Sequence Length | 2,048 | 8,192 |
| Trainable Module | VL-Connector | VL-Connector & LLM |
| Learning Rate | $1 \times 10^{-3}$ | $1 \times 10^{-5}$ |
| Batch Size | 512 | 256 |
| Warmup Ratio | 0.06 | 0.03 |

## F.3 TRAINING RECIPE

**Stage 1: Vision-language alignment.** We freeze most of the model's parameters and train only the vision-language connector on the Cambrian-Alignment-2.5M dataset (Tong et al., 2024a). Input images are padded to a fixed resolution of $384 \times 384$, and the maximum sequence length is set to 2048.

**Stage 2: Image instruction tuning.** We unfreeze both the vision-language connector and the LLM decoder, while keeping the vision encoder frozen. The model is then fine-tuned on the Cambrian-7M image instruction tuning dataset. Compared to Cambrian-1 (Tong et al., 2024a), we adopt the AnyRes strategy (Liu et al., 2024a) to enhance the model's image understanding capabilities. Specifically, input images are resized while preserving aspect ratio, then divided into multiple $384 \times 384$ sub-images. This enables the model to handle images with higher and more flexible resolutions. To accommodate the increased number of visual tokens introduced by the AnyRes strategy, we extend the sequence length to 8192. Detailed training configurations for stage 1 and 2 are provided in Table 9.

Table 10: **Training recipe for Cambrian-*S* stage 3 and stage 4.**

|  | Stage 3 (General Video Instruction Tuning) | Stage 4 (Spatial Video Instruction Tuning) |
|---|---|---|
| *Model* |  |  |
| Vision Encoder | SigLIP2-So400M | |
| Language Decoder | Qwen2.5-0.5B, 1.5B, 3B, 7B-Instruct | |
| VL-Connector | 2×MLP-GELU | |
| *Data Recipe* |  |  |
| Data Source | Cambrian-*S*-3M | VSI-590K + 590K general Video IT data (sampled from Cambrian-*S*-3M) |
| Video Frame Resolution | Pad (384×384) | Pad (384×384) |
| Frame Sampling Strategy | Uniform | Uniform |
| # Frames per Video | 64 | 128 |
| # Tokens per Video Frame | 64 | 64 |
| *Training Recipe* |  |  |
| Max Sequence Length | 8,192 | 16,384 |
| Trainable Modules | VL-Connector and LLM | |
| Learning Rate | $1 \times 10^{-5}$ | |
| Global Batch Size | 256 | |
| Warmup Ratio | 0.03 | |

**Stage 3: General video instruction tuning.**    To equip the model with general video understanding capabilities, we perform video instruction tuning on a mixture of curated Cambrian-*S*-3M video data and sampled image instruction data from Cambrian-7M. As in previous stages, the vision encoder remains frozen, and the remaining modules are fine-tuned. For image data, we reuse the sampling strategy from stage 2. For video data, we uniformly sample 64 frames per video, resize them to $384 \times 384$, and further downsample their feature maps to $8 \times 8$, *i.e.*, 64 tokens per frame.

**Stage 4: Spatial video instruction tuning.**    The final stage focuses on enhancing the model's spatial reasoning capabilities by fine-tuning on our proposed VSI-590K. To preserve general video and image understanding, we mixed 590K video samples from Cambrian-*S*-3M and 120K image samples from Cambrian-7M. Training settings are mostly consistent with stage 3, except for two key changes: (1) we increase the number of frames per video to 128, and (2) we extend the sequence length to 16,384, both to support richer temporal modeling. Detailed configurations for stage 3 and 4 are listed in Table 10.

### F.4    Infrastructure

All models in this paper are trained using TPU v4 Pods with the TorchXLA framework. To support large-scale video instruction tuning—where long sequence lengths introduce prohibitive computational and memory costs—we leverage GSPMD (Xu et al., 2021) and FlashAttention (Dao et al., 2022) implemented by Pallas.

GSPMD is an automatic parallelization system designed for flexible and user-friendly large-scale distributed training. It allows users to write training code as if for a single device, and then scale effortlessly across hundreds of devices with minimal changes. Our training framework is based on TorchXLA and GSPMD to shard data, model parameters, activations, and optimizer states across multiple devices. This reduces the peak memory usage and improves training throughput.

To accommodate long sequences, we integrate FlashAttention backed by Pallas, which significantly reduces TPU HBM (V-Mem) usage under long-context inputs. This enables us to scale the input sequence length up to 16,384 tokens for the 7B model on a TPU v4-512 Pod.

## G    Cambrian-*S* Additional Results

### G.1    Detailed Evaluation Setups

We describe the evaluation settings used for most image and video benchmarks, excluding VSI-Super. For image inputs, following the *any-resolution* design adopted in our training pipeline, each image is resized while preserving its aspect ratio, and its resolution is maximized so that it can be partitioned into at most nine 384×384 sub-images. For video inputs, we apply uniform frame sampling with a fixed number of frames. Specifically, checkpoints from stage 1 and stage 2 are evaluated with 32 uniformly sampled frames, while those from stage 3 and stage 4 use 64 and 128 frames, respectively.

### G.2    Post-training Recipe for Spatial Sensing

Table 11 studies the effectiveness of different post-training strategies. We find that stronger base models helps to achieve better SFT performance on spatial sensing tasks after finetuning on VSI-590K.

### G.3    VSI-Bench Sub-task Breakdown

Table 12 provides detailed breakdown of Gemini, GPT, and our Cambrian-*S* models' performance on VSI-Bench.

### G.4    Detailed Performance on Image and Video Benchmarks

Table 13 and Table 14 detail the performance of all our checkpoints (from stage 1 to stage 4 and from 0.5B to 7B) on image-based and video-based MLLM benchmarks, respectively. For image benchmarks, we report the results on MME (Yin et al., 2024), MMBench (Liu et al., 2024d), SeedBench (Li

Table 11: **Post-training exploration for spatial sensing.** We examine four base models with progressively increasing exposure to visual data, from image-only training to extensive video training, and analyze their distinct trends during spatial sensing tuning under two different data recipes. **A1**: only the connector is trained for image–language alignment; **A2**: A1 *w/.* Cambrian-7M image instruction-tuning data; **A3**: A2 further finetuned on 429K video instruction-tuning samples; **A4**: A2 further finetuned on 3M video instruction-tuning samples. From A1 to A4, the models show a monotonic improvement in video understanding ability. I-IT and V-IT denote instruction finetuning on image and video data, respectively. Finally, we show that stronger base models yield better SFT performance on spatial sensing tasks.

| Model | VSI-Bench | VideoMME | EgoSchema | Perception Test |
|---|---|---|---|---|
| *Different Base Models* | | | | |
| A1 (*w/o.* I-IT, *i.e.* QwenLM) | 21.4 | 44.2 | 42.9 | 44.5 |
| A2 (A1 + I-IT, *i.e.* Cambrian-1) | 25.8 | 53.7 | 48.1 | 55.4 |
| A3 (A2 + V-IT, 429K data) | 28.9 | 61.2 | 50.3 | 66.3 |
| A4 (A2 + V-IT, 3M data) | **35.7** | **62.6** | **77.0** | **70.9** |
| *SFT w/. VSI-590K* | | | | |
| from A1 | 57.2 | 40.3 | 38.7 | 52.3 |
| from A2 | 66.8 | 46.7 | 47.2 | 52.3 |
| from A3 | 68.8 | 52.3 | 48.4 | 55.8 |
| from A4 | **69.2** | **54.1** | **55.2** | **59.2** |
| *SFT w/. VSI-590K & general V-IT data mixture* | | | | |
| from A1 | 61.3 | 60.5 | 52.8 | 65.0 |
| from A2 | 63.2 | **62.6** | 52.9 | 65.6 |
| from A3 | 64.0 | 61.0 | 54.9 | 66.8 |
| from A4 | **65.1** | 61.9 | **77.3** | **71.2** |

Table 12: **VSI-Bench sub-task breakdown.** Best results are **bolded**. Notably, even without any route planning data in training, Cambrian-*S*-7B outperforms Gemini-1.5-Pro on this task.

| Methods | Avg. | Obj. Count | Abs. Dist. | Obj. Size | Room Size | Rel. Dist. | Rel. Dir. | Route Plan | Appr. Order |
|---|---|---|---|---|---|---|---|---|---|
| | | Numerical Answer | | | | Multiple-Choice Answer | | | |
| *Statistics* | | | | | | | | | |
| Chance Level (Random) | - | - | - | - | - | 25.0 | 36.1 | 28.3 | 25.0 |
| Chance Level (Frequency) | 34.0 | 62.1 | 32.0 | 29.9 | 33.1 | 25.1 | 47.9 | 28.4 | 25.2 |
| *Proprietary Models (API)* | | | | | | | | | |
| GPT-4o | 34.0 | 46.2 | 5.3 | 43.8 | 38.2 | 37.0 | 41.3 | 31.5 | 28.5 |
| Gemini-1.5 Flash | 42.1 | 49.8 | 30.8 | 53.5 | 54.4 | 37.7 | 41.0 | 31.5 | 37.8 |
| Gemini-1.5 Pro | 45.4 | 56.2 | 30.9 | 64.1 | 43.6 | 51.3 | 46.3 | 36.0 | 34.6 |
| Gemini-2.5 Pro | 51.5 | 43.8 | 34.9 | 64.3 | 42.8 | 61.1 | 47.8 | **45.9** | 71.3 |
| *Open-source Models* | | | | | | | | | |
| Cambrian-*S*-7B | **67.5** | **73.2** | **50.5** | **74.9** | **72.2** | **71.1** | **76.2** | 41.8 | **80.1** |
| Cambrian-*S*-3B | 57.3 | 70.7 | 40.6 | 68.0 | 46.3 | 64.8 | 61.9 | 27.3 | 78.8 |
| Cambrian-*S*-1.5B | 54.8 | 68.4 | 40.0 | 61.5 | 50.1 | 62.4 | 48.9 | 29.9 | 77.5 |
| Cambrian-*S*-0.5B | 50.6 | 67.9 | 35.4 | 52.2 | 52.5 | 52.3 | 46.5 | 25.8 | 72.2 |

et al., 2024a), GQA (Hudson & Manning, 2019), ScienceQA (Saikh et al., 2022), MMMU (Yue et al., 2024), MathVista (Lu et al., 2024), AI2D (Kembhavi et al., 2016), ChartQA (Masry et al., 2022), OCRBench (Liu et al., 2024e), TextVQA (Zhou et al., 2018), DocVQA (Mathew et al., 2021), MMVP (Tong et al., 2024b), RealworldQA (xAI, 2024), and CVBench (Tong et al., 2024a), following Cambrian-1's grouping strategy.

## G.5 Contributions from Image-based and Video-based Instruction Tuning

To elaborate on the respective contributions of image-based and video-based instruction tuning to a model's final video understanding capabilities, we conducted a series of experiments. These experiments employed varying proportions of image and video data during the finetuning stages, and we observed the resulting performance trends across diverse video benchmarks.

More specifically, for the initial image MLLM training, we randomly sampled 1M, 4M, and 7M image question-answering (QA) pairs from Cambrian-7M to train distinct models. Subsequently, for video-specific finetuning, we randomly sampled 25%, 50%, 75%, and 100% of video QA pairs from LLaVA-Video-178K (∼1.6M data samples in total) to perform video-only finetuning on each of these

Table 13: **Detailed results of Cambrian-S checkpoints on image MLLM benchmarks.**

| Method | General Avg | MME$^P$ | MMB | SEED$^I$ | GQA | Knowledge Avg | SQA$^I$ | MMMU$^V$ | MathVista$^M$ | AI2D | OCR & Chart Avg | ChartQA | OCRBench | TextVQA | DocVQA | Vision-Centric Avg | MMVP | RealworldQA | CV-Bench$^{2D}$ | CV-Bench$^{3D}$ |
|---|---|---|---|---|---|---|---|---|---|---|---|---|---|---|---|---|---|---|---|---|
| *Open-source Models* | | | | | | | | | | | | | | | | | | | | |
| Mini-Gemini-HD-8B | 72.7 | 1606.0 | 72.7 | 73.2 | 64.5 | 55.7 | 75.1 | 37.3 | 37.0 | 73.5 | 62.9 | 59.1 | 47.7 | 70.2 | 74.6 | 51.5 | 18.7 | 62.1 | 62.2 | 63.0 |
| LLaVA-NeXT-8B | 72.5 | 1603.7 | 72.1 | 72.7 | 65.2 | 55.6 | 72.8 | 41.7 | 36.3 | 71.6 | 63.9 | 69.5 | 49.0 | 64.6 | 72.6 | 56.6 | 38.7 | 60.1 | 62.2 | 65.3 |
| Cambrian-1-8B | 73.1 | 1,547.1 | 75.9 | 74.7 | 64.6 | 61.3 | 80.4 | 42.7 | 49.0 | 73.0 | 71.3 | 73.3 | 62.4 | 71.7 | 77.8 | 65.0 | 51.3 | 64.2 | 72.3 | 72.0 |
| *Cambrian-S-7B* | | | | | | | | | | | | | | | | | | | | |
| Stage 1 | 11.5 | 209.9 | 29.6 | 5.6 | 0.1 | 2.5 | 3.1 | 2.2 | 2.9 | 1.7 | 7.4 | 0.9 | 27.6 | 0.9 | 0.1 | 0.9 | 0.0 | 2.7 | 0.8 | 0.0 |
| Stage 2 | 74.9 | 1604.6 | 79.0 | 76.3 | 64.0 | 63.9 | 83.7 | 48.7 | 45.3 | 78.1 | 79.1 | 78.9 | 67.6 | 79.2 | 90.6 | 66.3 | 53.3 | 67.7 | 70.0 | 74.0 |
| Stage 3 | 74.4 | 1583.9 | 79.7 | 76.4 | 62.4 | 60.4 | 82.2 | 46.2 | 36.1 | 77.0 | 75.5 | 75.3 | 64.0 | 77.1 | 85.6 | 67.0 | 58.0 | 66.1 | 71.8 | 72.3 |
| Stage 4 | 74.8 | 1598.4 | 80.4 | 77.0 | 61.8 | 64.6 | 82.7 | 48.0 | 50.6 | 76.9 | 75.2 | 74.7 | 64.8 | 76.6 | 84.8 | 70.5 | 60.0 | 64.8 | 74.3 | 83.0 |
| *Cambrian-S-3B* | | | | | | | | | | | | | | | | | | | | |
| Stage 1 | 8.3 | 9.3 | 31.7 | 1.0 | 0.0 | 0.9 | 0.9 | 0.9 | 0.1 | 1.7 | 7.0 | 0.0 | 28.1 | 0.0 | 0.1 | 0.7 | 0.0 | 2.1 | 0.7 | 0.0 |
| Stage 2 | 71.9 | 1524.6 | 74.8 | 74.2 | 62.1 | 55.5 | 78.7 | 42.8 | 27.8 | 72.7 | 72.0 | 69.8 | 63.9 | 71.5 | 82.7 | 59.0 | 37.3 | 62.4 | 65.6 | 70.7 |
| Stage 3 | 72.1 | 1495.7 | 76.5 | 75.1 | 61.8 | 58.5 | 79.4 | 42.2 | 41.3 | 71.2 | 69.6 | 68.0 | 61.3 | 69.6 | 79.4 | 62.5 | 46.0 | 61.2 | 70.6 | 72.4 |
| Stage 4 | 71.5 | 1485.6 | 76.0 | 75.1 | 60.8 | 58.7 | 78.7 | 42.1 | 43.0 | 70.9 | 69.6 | 70.0 | 60.5 | 68.7 | 79.1 | 65.6 | 50.0 | 60.1 | 76.1 | 76.3 |
| *Cambrian-S-1.5B* | | | | | | | | | | | | | | | | | | | | |
| Stage 1 | 11.7 | 282.1 | 28.6 | 0.8 | 3.2 | 3.8 | 6.9 | 4.2 | 1.4 | 2.6 | 7.9 | 1.0 | 27.8 | 1.4 | 1.5 | 0.7 | 0.0 | 0.0 | 2.9 | 0.0 |
| Stage 2 | 68.5 | 1417.3 | 71.3 | 71.2 | 60.6 | 50.9 | 75.5 | 41.1 | 20.8 | 66.1 | 68.0 | 64.8 | 59.9 | 68.8 | 78.6 | 54.4 | 39.3 | 59.7 | 60.3 | 58.3 |
| Stage 3 | 68.1 | 1423.2 | 70.5 | 72.1 | 58.7 | 52.6 | 72.4 | 40.8 | 32.3 | 64.8 | 64.2 | 59.5 | 57.6 | 66.7 | 72.9 | 54.6 | 40.0 | 59.9 | 60.7 | 57.8 |
| Stage 4 | 68.0 | 1394.4 | 70.1 | 73.5 | 58.7 | 54.7 | 72.3 | 42.0 | 39.7 | 64.7 | 65.6 | 63.1 | 58.0 | 66.6 | 74.8 | 59.2 | 43.3 | 54.5 | 62.6 | 76.3 |
| *Cambrian-S-0.5B* | | | | | | | | | | | | | | | | | | | | |
| Stage 1 | 10.1 | 379.6 | 10.7 | 9.0 | 1.8 | 6.2 | 8.4 | 8.9 | 1.9 | 5.5 | 3.0 | 0.2 | 7.9 | 2.0 | 1.9 | 10.9 | 0.7 | 10.6 | 20.1 | 12.3 |
| Stage 2 | 57.7 | 1124.3 | 56.6 | 61.7 | 56.1 | 38.6 | 61.5 | 31.0 | 10.5 | 51.5 | 56.0 | 51.1 | 51.0 | 58.7 | 63.1 | 41.2 | 23.3 | 51.8 | 45.6 | 44.1 |
| Stage 3 | 58.6 | 1200.0 | 55.8 | 63.5 | 55.3 | 41.2 | 62.7 | 32.6 | 18.0 | 51.4 | 52.1 | 46.6 | 46.8 | 56.0 | 59.1 | 45.5 | 22.0 | 52.8 | 52.2 | 54.9 |
| Stage 4 | 60.0 | 1190.8 | 60.7 | 66.4 | 53.5 | 44.0 | 63.4 | 34.0 | 28.6 | 50.1 | 52.6 | 48.0 | 47.1 | 56.6 | 58.6 | 48.7 | 26.0 | 51.1 | 51.6 | 66.2 |

Table 14: **Detailed results of Cambrian-S checkpoints on video MLLM benchmarks.**

| Model | Base LLM | VSI-Bench | Tomato | HourVideo | Video$^{MME}$ | EgoSchema | Video$^{MMMU}$ | LongVBench | MVBench | Percept. Test |
|---|---|---|---|---|---|---|---|---|---|---|
| Cambrian-S-7B | | | | | | | | | | |
| Stage 1 | | 21.4 | 21.0 | 27.5 | 44.3 | 42.9 | 11.3 | 32.3 | 43.9 | 44.4 |
| Stage 2 | Qwen2.5-7B | 24.6 | 20.1 | 31.3 | 52.3 | 47.5 | 28.1 | 51.1 | 49.2 | 53.5 |
| Stage 3 | | 35.7 | 30.3 | 38.9 | 62.8 | 76.9 | 38.3 | 56.7 | 66.3 | 70.8 |
| Stage 4 | | 67.5 | 27.9 | 36.5 | 63.3 | 76.3 | 38.3 | 59.4 | 64.8 | 69.8 |
| Cambrian-S-3B | | | | | | | | | | |
| Stage 1 | | 0.7 | 16.5 | 0.7 | 15.9 | 19.5 | 8.4 | 23.8 | 30.6 | 18.6 |
| Stage 2 | Qwen2.5-3B | 22.3 | 21.3 | 31.7 | 49.4 | 42.2 | 26.0 | 48.7 | 44.5 | 47.0 |
| Stage 3 | | 23.3 | 26.3 | 35.9 | 58.9 | 73.4 | 27.1 | 52.0 | 61.0 | 65.7 |
| Stage 4 | | 57.3 | 26.0 | 36.8 | 60.1 | 73.6 | 26.3 | 52.3 | 60.2 | 65.9 |
| Cambrian-S-1.5B | | | | | | | | | | |
| Stage 1 | | 21.1 | 23.5 | 26.2 | 40.1 | 33.0 | 18.7 | 38.5 | 40.8 | 45.2 |
| Stage 2 | Qwen2.5-1.5B | 22.6 | 24.6 | 34.4 | 47.8 | 38.2 | 20.7 | 46.9 | 45.3 | 49.8 |
| Stage 3 | | 23.4 | 23.1 | 33.2 | 56.1 | 67.8 | 28.6 | 49.4 | 58.2 | 63.6 |
| Stage 4 | | 54.8 | 22.2 | 31.2 | 56.4 | 69.0 | 25.0 | 50.2 | 57.1 | 63.2 |
| Cambrian-S-0.5B | | | | | | | | | | |
| Stage 1 | | 16.7 | 23.6 | 23.4 | 26.4 | 21.5 | 13.1 | 25.0 | 34.3 | 37.0 |
| Stage 2 | Qwen2.5-0.5B | 19.6 | 20.0 | 27.9 | 37.4 | 29.7 | 17.3 | 39.0 | 40.2 | 46.3 |
| Stage 3 | | 18.8 | 23.9 | 29.5 | 41.8 | 63.8 | 16.7 | 44.9 | 50.7 | 56.1 |
| Stage 4 | | 50.4 | 23.8 | 28.1 | 44.0 | 62.4 | 15.9 | 43.8 | 51.8 | 56.0 |

pretrained image MLLMs. The hyperparameters for image instruction tuning and video finetuning were maintained as detailed in Table 9 and Table 10, respectively. The experimental results, presented in Table 15, yield the following observations:

- *Models trained with more image data do not inherently outperform those trained with less when evaluated on video benchmarks without finetuning.* As indicated in the table, direct evaluation on video benchmarks reveals comparable performance across all three models, which were initially trained on 1M, 4M, and 7M image datasets, respectively.

- *Finetuning on video data can be generally beneficial for models pretrained with larger image datasets, though not universally.* When all models were finetuned on 100% video data, the model initially trained on 7M images outperformed the other two on 5 out of 9 video benchmarks (specifically, HourVideo, VideoMME, EgoSchema, LongVideoBench, and Perception Test).

- *Incorporating video data into the training process consistently benefits performance across all video benchmarks.* We observed that finetuning an image-based MLLM with video

Table 15: **Video MLLM performance trained with different proportions of image and video data.**

| Image data | Video data | VSI-Bench | Tomato | HourVideo | VideoMME | EgoSchema | VideoMMMU | LongVBench | MVBench | Percept. Test |
|---|---|---|---|---|---|---|---|---|---|---|
| *Chance-Level* | - | 34.0 | 22.0 | 20.0 | 25.0 | 20.0 | 14.0 | 25.0 | 27.3 | 33.3 |
| | 0% | 26.0 | 20.2 | 32.5 | 52.1 | 46.9 | 32.0 | 51.4 | 50.5 | 54.2 |
| | 25% | 32.4 | 25.4 | 36.2 | 60.4 | 47.0 | 40.1 | 53.5 | 57.0 | 61.9 |
| 1M | 50% | 33.3 | 27.2 | 36.2 | 61.7 | 47.1 | 40.1 | 53.2 | 59.2 | 64.3 |
| | 75% | 32.7 | 28.8 | 34.4 | 60.7 | 48.7 | 37.7 | 53.3 | 59.5 | 66.3 |
| | 100% | 34.4 | 28.4 | 35.1 | 61.3 | 48.9 | 39.6 | 53.0 | 60.1 | 67.5 |
| | 0% | 26.7 | 20.5 | 31.8 | 53.1 | 44.8 | 32.0 | 52.1 | 51.5 | 54.9 |
| | 25% | 32.3 | 26.7 | 37.0 | 61.3 | 45.0 | 38.6 | 53.1 | 57.6 | 61.9 |
| 4M | 50% | 31.9 | 27.4 | 37.2 | 61.9 | 45.7 | 38.1 | 54.2 | 59.5 | 65.2 |
| | 75% | 33.8 | 27.9 | 36.2 | 61.1 | 47.3 | 40.9 | 53.1 | 60.1 | 67.0 |
| | 100% | 33.8 | 28.0 | 35.5 | 60.5 | 50.2 | 40.2 | 52.2 | 60.5 | 67.7 |
| | 0% | 25.8 | 18.9 | 31.6 | 53.7 | 48.1 | 31.9 | 52.5 | 51.4 | 55.4 |
| | 25% | 31.5 | 24.6 | 36.7 | 61.3 | 48.8 | 37.7 | 54.7 | 58.3 | 62.3 |
| 7M | 50% | 31.4 | 27.6 | 36.6 | 61.0 | 49.0 | 37.9 | 53.6 | 59.7 | 65.6 |
| | 75% | 31.8 | 27.0 | 35.7 | 61.8 | 50.7 | 38.0 | 53.0 | 60.2 | 67.9 |
| | 100% | 32.6 | 27.7 | 37.3 | 62.1 | 52.4 | 39.4 | 54.3 | 60.6 | 68.8 |

data, even a small portion such as 25%, improved its performance on all evaluated video benchmarks.

- *Increasing the amount of video data used for finetuning does not guarantee consistent performance improvements across all benchmarks.* While video finetuning is generally advantageous, some benchmarks (*e.g.*, VideoMME, VSI-Bench, Tomato) do not show further gains with more video data. For instance, models finetuned with 100% video data exhibited performance on par with those finetuned with only 25% video data on the VideoMME benchmark. Only EgoSchema, MVBench, and Perception Test demonstrated consistent benefits from increased video data, a phenomenon we hypothesize is related to the underlying video distribution of the training videos.

## G.6 ON THE TRADE-OFF BETWEEN SPATIAL SENSING AND GENERAL VIDEO UNDERSTANDING

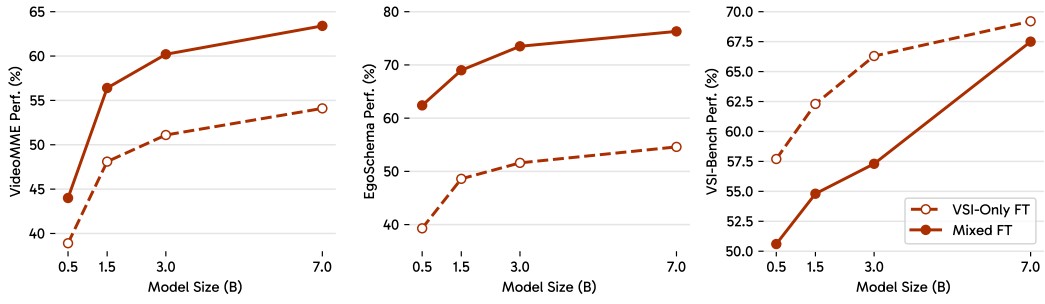

Figure 16: **On the trade-off between spatial-sensing and general video understanding.**

In Section 3.3, we compare model performance when fine-tuned either on VSI-590K alone or on a mixture of VSI-590K and general video data. We observe that fine-tuning on VSI-590K alone consistently yields higher performance on spatial sensing tasks, whereas mixed-data fine-tuning offers a better balance between spatial sensing and general video understanding. To further explore this trade-off across model scales, we conduct fine-tuning after stage 3 using either VSI-590K alone or the mixed dataset, under four different model sizes: 0.5B, 1B, 3B, and 7B parameters. We then evaluate these models on both general video understanding and spatial sensing benchmarks, as shown in Fig. 16.

The results confirm that the previous conclusion holds across all scales: VSI-590K-only fine-tuning excels at spatial sensing, while mixed-data fine-tuning provides a better overall balance. Notably, however, the performance gap on VSI-Bench narrows as model size increases. We attribute this to the greater capacity of larger models to learn and retain diverse capabilities. This trend suggests that scaling to even larger models may further mitigate the spatial sensing performance drop typically observed when fine-tuning with mixed data.

# H  PREDICTIVE SENSING

## H.1  LATENT FRAME PREDICTION IMPLEMENTATION DETAILS

**Latent frame prediction head.**  As shown in Algorithm 2, our next-frame prediction head is a simple two-layer MLP with GELU activation (Hendrycks, 2016), running in parallel with the MLLM's original language model head. The output dimension is set to 1152, matching the output dimension of our vision encoder (*i.e.*, `siglip2-so400m-patch14-384`).

**Algorithm 2: Latent frame prediction (LFP) head architecture (in PyTorch style).**

```
LFPHead(
  Sequential(
    (0): Linear(in_features=3584, out_features=3584, bias=True)
    (1): GELU(approximate=none)
    (2): Linear(in_features=3584, out_features=1152, bias=True)
    )
)
```

**On the balance between LFP and instruction tuning losses.**  As mentioned in Section 4.1, to build the model's internal world model, we slightly modify our stage 4, introducing two auxiliary losses (*i.e.*, cosine distance and mean-squared error) to optimize the next frame prediction objectiveness. A coefficient is applied to balance the LFP loss against the instruction tuning loss, which we ablate in Table 16.

Table 16: **Evaluation results across different benchmarks with varying LFP loss weights.** Our default setup (0.1 loss coefficient) is highlighted in  gray .

| LFP loss coeffcient | VSI-Bench | VideoMME | EgoSchema | Perception Test |
|---|---|---|---|---|
| 0.0 (i.e., *No LFP Loss*) | 67.5 | 63.4 | 76.8 | 69.9 |
| 0.1 | 66.1 | 63.9 | 76.9 | 69.7 |
| 0.5 | 60.8 | 63.6 | 77.2 | 66.4 |
| 1.0 | 56.6 | 61.0 | 72.9 | 65.1 |

## H.2  MEMORY FRAMEWORK DESIGN FOR VSI-SUPER RECALL

As introduced in main paper (and shown in Algorithm 3), our predictive memory mechanism comprises three distinct memory levels ($M_s$, $M_l$, $M_w$) and four key transition functions governing their interaction: *Sensory Streaming*, *Memory Compression*, *Memory Consolidation*, and *Retrieval*. This section details the implementation of these functions.

**Basic memory units.**  For our implementation, we utilize the *encoded key-value pairs* from each Large Language Model (LLM) layer as the basic memory units. This choice, rather than using output latent features from a vision encoder or vision-language connector, allows us to fully leverage the LLM's internal capabilities for memory construction without requiring external modules. This design decision will be elaborated upon in subsequent sections.

**Streaming sensing.**  Each incoming frame is initially processed independently by the vision encoder and the vision-language connector with a window size of $W_s$. Subsequently, it is further encoded by

the LLM, referencing selected previous frames. The key-value pairs from these preceding frames, cached in the *Sensory memory buffer* ($M_s$), provide the necessary context for this encoding step.

**Surprise-based memory compression.** In the meantime of encoding a single frame, we assess its "surprise" level. This is achieved by calculating the difference between the model's prediction for the current frame and the actual ground truth observation (both in the latent feature space). When a frame of timestamp $t$ is moved from the sensory memory buffer $M_s$ to the long-term memory $M_l$, if it is deemed non-surprising (*i.e.*, its surprise score is below a predefined threshold $T_s$), we will downsample its' key-value pairs by a factor of 2 along the spatial ($H \times W$) dimension. This surprise-based compression mitigates redundancy in the information stored within $M_l$.

**Surprise-based memory consolidation.** Long-term memory $M_l$ is initialized with a predefined budget size $B_{long}$ (*e.g.*, 32,768 tokens). When the volume of memory tokens surpasses this budget, we apply a *surprise-based* consolidation function to $M_l$ to ensure it remains within the allocated limit. Our consolidation function is straightforward yet effective: we identify the surprise score associated with each frame in $M_l$. Then, the frame with the lowest surprise score is removed (or "forgotten"). Then, we merge or drop some of these frames according to their surprise scores (we tried three different strategies here: 1. forget the oldest memory, 2. forget the least surprise memory, and 3. forget the least surprise memory while merging adjacent surprise memories if any adjacent surprise memories exist). This process is iterated until the total size of $M_l$ falls below the budget.

**Retrieval.** Upon receiving a user query $q$, we first retrieve the most relevant frames from the long-term memory ($M_l$) to construct the working memory ($M_w$). This $M_w$ then serves as the context for answering the user's query. To perform this retrieval efficiently without resorting to external modules, we utilize the inherent similarity measurement capabilities of the LLM's attention mechanism. Specifically, for each transformer layer, the user query $q$ is transformed into the attention mechanism's query feature space. We then compute the similarity between this query feature and the key features of each frame stored in $M_l$. Similarity is measured using cosine distance, and for simplicity, multi-head features are treated as a single feature. The $k$ frames with the highest similarity scores have their key-value pairs selected and utilized by the attention mechanism to further encode the user query.

---

**Algorithm 3: Memory framework design for VSI-SUPER Recall.**

**Input:** Frames $\{f_1, \ldots, f_T\}$, User query $q$
**Input:** Encoder $\mathcal{E}$, Decoder $\mathcal{D}$, Surprise Estimator $\mathcal{S}$, Surprise threshold $\tau$
**Input:** Compression function $\mathcal{C}$, Consolidation function $\mathcal{G}$, Retrieval function $\mathcal{R}$
**Input:** Sensory memory $\mathcal{M}_s \leftarrow \emptyset$ with budget $B_s$, Long-term memory $\mathcal{M}_l \leftarrow \emptyset$ with budget $B_l$, Working memory $\mathcal{M}_w \leftarrow \emptyset$

1 **for** $t \leftarrow 1$ **to** $T$ **do**
2 $\quad z_t \leftarrow \mathcal{E}(f_t, \mathcal{M}_s)$;
3 $\quad \mathcal{M}_s \leftarrow \mathcal{M}_s \cup \{z_t\}$ ;            `// Streaming sensing`
4 $\quad s_t \leftarrow \mathcal{S}(f_t, \mathcal{M}_s)$ ;            `// Surprise estimation`
5 $\quad$ **while** $|\mathcal{M}_s| > B_s$ **do**
6 $\quad\quad$ Dequeue $z_{\text{old}}$ from $\mathcal{M}_s$;
7 $\quad\quad m \leftarrow \mathbf{1}[s_t \geq \tau] \cdot z_{\text{old}} + \mathbf{1}[s_t < \tau] \cdot \mathcal{C}(z_{\text{old}})$ ;    `// Selective compression`
8 $\quad\quad \mathcal{M}_l \leftarrow \mathcal{M}_l \cup \{m\}$;
9 $\quad\quad$ **if** $|\mathcal{M}_l| > B_l$ **then**
10 $\quad\quad\quad \mathcal{M}_l \leftarrow \mathcal{G}(\mathcal{M}_l)$ ;            `// Memory consolidation`

11 $\mathcal{M}_w \leftarrow \mathcal{R}(q, \mathcal{M}_l)$ ;            `// Retrieve working memory`
12 $\hat{a} \leftarrow \mathcal{D}(q, \mathcal{M}_w)$ ;            `// Answering query with` $\mathcal{M}_w$
13 **return** $\hat{a}$

---

### H.3 AGENTIC FRAMEWORK DESIGN FOR VSI-SUPER COUNT

Algorithm 4 presents our agentic framework for the VSI-SUPER Count task. Similar to the memory design in Algorithm 3, we encode sensory frames using a sliding window approach with a window size of $W_s$. The latent frame prediction module continuously estimates the expected next frame and computes the prediction error to quantify how "surprise" the actual next frame is. As new frame arrivs,

the oldest frames that exceed the sensory memory window are dequeued and stored in the long-term memory. If a dequeued frame is deemed "surprising" (*i.e.*, its prediction error exceeds a predefined threshold $\tau$), which may indicate a scene or spatial boundary, we trigger a query response using the accumulated long-term memory and reset it afterward. The generated response is then stored in the answer memory bank. The final answer is computed as the aggregation of all intermediate answers stored in this bank.

---

**Algorithm 4: Agentic framework design for VSI-SUPER Count task.**

**Input:** Frames $\{f_1, \ldots, f_T\}$, user query $q$
**Input:** Encoder $\mathcal{E}$, Decoder $\mathcal{D}$, Surprise Estimator $\mathcal{S}$, threshold $\tau$
**Input:** Sensory memory $\mathcal{M}_s \leftarrow \emptyset$ with budget $B_s$
**Input:** Long-term memory $\mathcal{M}_l \leftarrow \emptyset$, Answer memory bank $\mathcal{M}_{\text{Ans}} \leftarrow \emptyset$

1   **for** $t \leftarrow 1$ **to** $T$ **do**
2     $z_t \leftarrow \mathcal{E}(f_t, \mathcal{M}_s)$;
3     $\mathcal{M}_s \leftarrow \mathcal{M}_s \cup \{z_t\}$ ;                // Streaming sensing
4     $s_t \leftarrow \mathcal{S}(f_t, \mathcal{M}_s)$ ;               // Surprise estimation
5     **if** $|\mathcal{M}_s| > B_s$ **then**
6       Remove oldest $z_{\text{old}}$ from $\mathcal{M}_s$;
7       $\mathcal{M}_l \leftarrow \mathcal{M}_l \cup \{z_{\text{old}}\}$ ;        // Store to long-term memory
8     **if** $s_t \geq \tau$ **then**
9       $\hat{a} \leftarrow \mathcal{D}(q, \mathcal{M}_l)$ ;      // Answer query using long-term memory
10      $\mathcal{M}_{\text{Ans}} \leftarrow \mathcal{M}_{\text{Ans}} \cup \{\hat{a}\}$;
11      $\mathcal{M}_l \leftarrow \emptyset$ ;                 // Reset long-term memory

12 **return** $\text{Sum}(\mathcal{M}_{\text{Ans}})$

---

### H.4 COMPARISONS WITH EXISTING LONG-VIDEO METHODS

We compare our method (both surprise-driven memory and agentic framework) with existing methods designed for long-video understanding, in Table 17. Specifically, all experiments here are conducted with our LFP-finetuned Cambrian-*S*-7B, with a different strategy to handle the ever-expanding visual sensory input. For MovieChat, we follow the official implementation in (Song et al., 2024), maintain a fixed-size long-term memory bank, and set the long-term and short-term memory budgets to 64 and 16, respectively. For Flash-VStream (Zhang et al., 2025a), as its abstract memory module introduces additional parameters and requires a dedicated training process, we only implement the three remaining memory components (*i.e.*, spatial memory, temporal memory, and retrieved memory), and keeping all other hyperparameters aligned with the default setup.

Table 17: **Compare our framework with existing long-video methods on VSI-SUPER.**

| Eval Setups | VSR (Duration in Mins.) | | | | | VSC (Duration in Mins.) | | | |
|---|---|---|---|---|---|---|---|---|---|
| | 10 | 30 | 60 | 120 | 240 | 10 | 30 | 60 | 120 |
| MovieChat | 18.3 | 21.7 | 16.7 | 26.7 | 25.6 | 0.0 | 0.0 | 0.0 | 0.0 |
| Flash-VStream | 28.3 | 33.3 | 23.3 | 28.3 | 31.7 | 0.0 | 0.0 | 0.0 | 0.0 |
| Ours | 45.0 | 41.7 | 40.0 | 40.0 | 40.0 | 40.6 | 42.0 | 35.0 | 34.0 |

### H.5 VISUALIZATION OF CAMBRIAN-*S*'S PREDICTIONS *v.s.* GROUND TRUTH ON VSC

Fig. 17 visualizes the predicted counts on VSC, revealing that Cambrian-*S*'s predictions exhibit strong linear growth with the ground truth, aligning well with the $y = x$ perfect-count line (gray dashed). Conversely, Gemini-2.5-Flash's predictions are confined to a limited range of small values and do not scale as the actual count increases, highlighting its poor extrapolation capabilities.

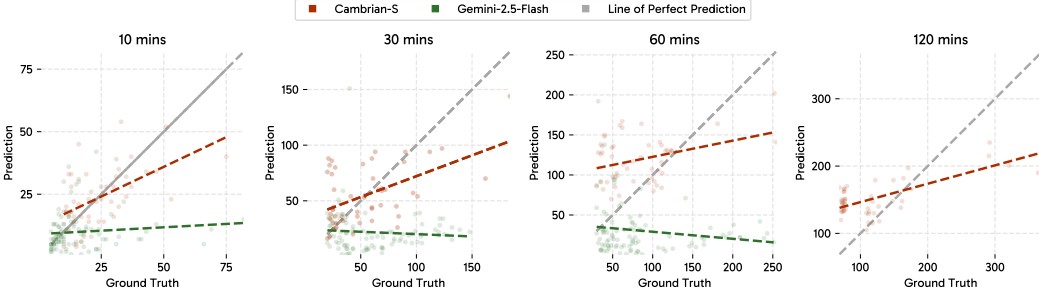

Figure 17: **Cambrian-*S* scales to higher ground truth object counts whereas Gemini saturates.** Predicted counts are plotted against ground-truth counts for videos of different lengths (10, 30, 60, and 120 minutes). Using surprise-driven segmentation, Cambrian-*S*'s predicted counts grow approximately linearly with the ground-truth, tracking the $y = x$ perfect-count line (gray dashed), whereas Gemini-2.5-Flash's predicted counts remain clustered near small values and fail to increase with ground-truth count, indicating early saturation and poor extrapolation to larger counts.

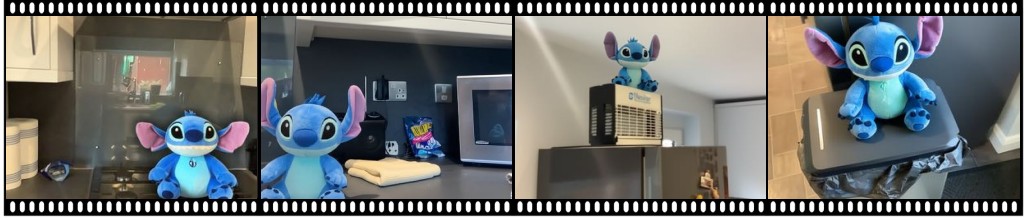

Which of the following correctly represents the order in which the Stitch appeared in the video?
A. Stove, Trash bin, Refrigerator, Counter    B. Trash bin, Refrigerator, Counter, Stove
C. Stove, Counter, Refrigerator, Trash bin    D. Trash bin, Stove, Counter, Refrigerator

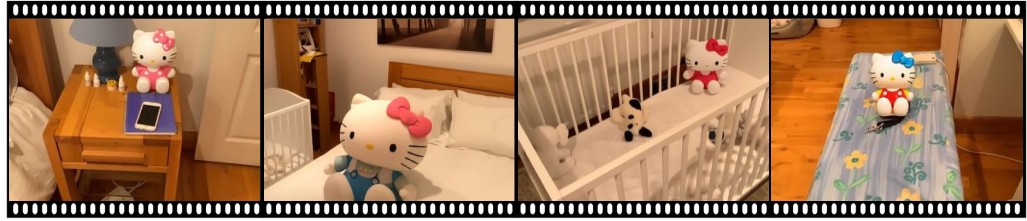

Which of the following correctly represents the order in which the Hello Kitty appeared in the video?
A. Nightstand, Bed, Crib, Blue bench    B. Blue bench, Crib, Nightstand, Bed
C. Bed, Nightstand, Blue bench, Crib    D. Blue bench, Bed, Crib, Nightstand

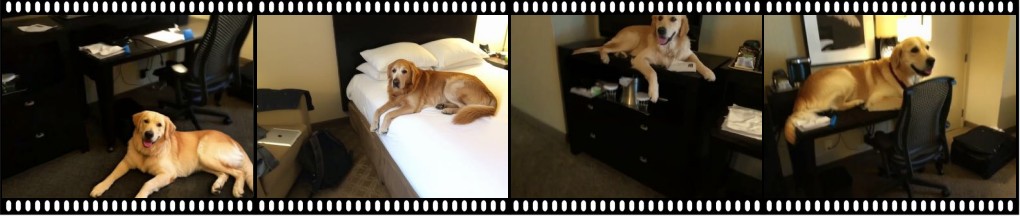

Which of the following correctly represents the order in which the Golden Retriever appeared in the video?
A. Bed, Table, Chest of drawers, Floor    B. Table, Chest of drawers, Bed, Floor
C. Chest of drawers, Floor, Table, Bed    D. Floor, Bed, Chest of drawers, Table

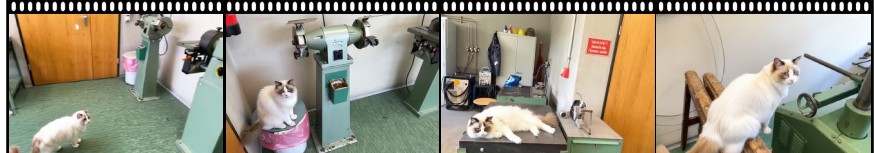

Which of the following correctly represents the order in which the white Ragdoll cat appeared in the video?
A. Ground, Trash bin, Bench, Table    B. Table, Bench, Ground, Trash bin
C. Ground, Trash bin, Table, Bench    D. Trash bin, Bench, Table, Ground

Figure 18: More examples of our VSI-SUPER Recall benchmark. Note that only edited frames are visualized.

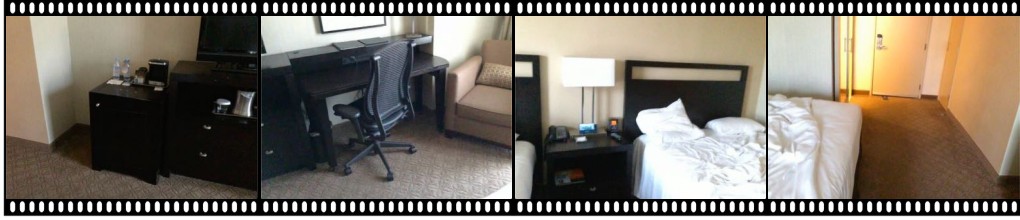

**Absolute Direction (Object)**

Standing by the backpack, looking toward the table, how far counterclockwise in degrees must I turn to see the trash bin?
Answer: 334.09

**Absolute Distance**

Measuring from the closest points of each, how far apart are the chair and the door in meters?
Answer: 2.32

**Absolute Distance**

Considering the chair and the door, which object's longest edge is the shorter?
A. Door
B. Chair

Figure 19: Examples of VSI-590K (Annotated Real Video).

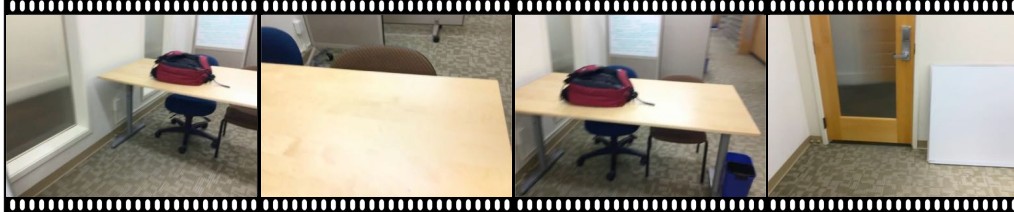

**Object Appearance Order**

Determine the initial appearance order of these categories in the video: door, chair, lamp, refrigerator.
A. refrigerator, door, lamp, chair            B. refrigerator, chair, door, lamp
C. refrigerator, chair, lamp, door            D. door, chair, lamp, refrigerator

**Absolute Size**

Provide the longest side's length for the door in inches.
Answer: 72.00

**Room Size**

Indicate the room's dimensions in square feet. If there's more than one room, estimate their total size.
Answer: 232.76

Figure 20: Examples of VSI-590K (Annotated Real Video).

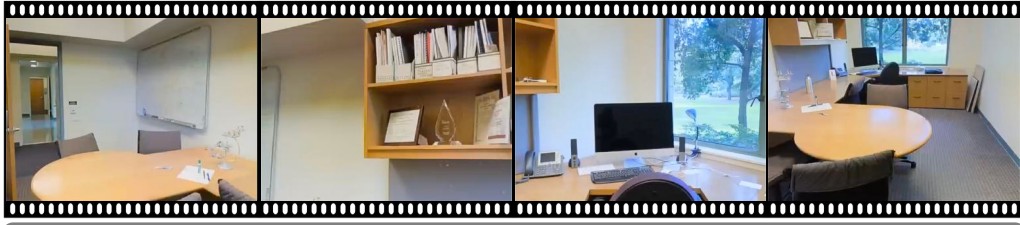

Relative Direction (Object Perspective)

Facing the door while standing near the window, in which of the following positions is the board relative to me: front-left, front-right, back-left, or back-right? Use Cartesian quadrants, with me at the origin looking toward positive y-axis
A. Back-right
B. Front-right
C. Front-left
D. Back-left

Relative Distance (Object Perspective)

Identify the object among (bookcase, chair, board, door) that is closest to the window based on the shortest distance between their closest points. Choose the nearest instance if several exist.
A. Bookcase
B. Chair
C. Board
D. Door

Figure 21: Examples of VSI-590K (Annotated Real Video).

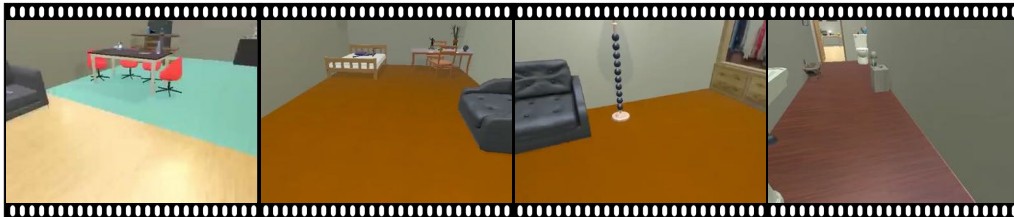

Relative Distance (Object Perspective)

If I am standing by the dresser and facing the chair, is the closet to my left, right, or back? An object is to my back if I would have to turn at least 135 degrees in order to face it.
A. Left
B. Right
C. Back

Figure 22: Examples of VSI-590K (Annotated Simulated Video).

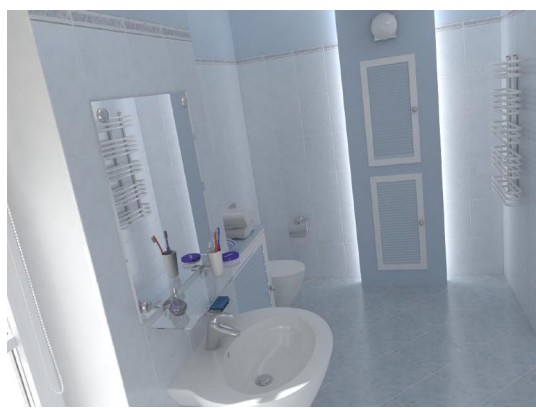

---

**Relative Direction (Object Perspective)**

With the toilet beside me and facing the cabinet, is the lamp positioned front-left, front-right, back-left, or back-right relative to me, based on Cartesian plane quadrants?
A. Back-right
B. Front-right
C. Front-left
D. Back-left

---

**Relative Distance (Object Perspective)**

Identify the object among (bookcase, chair, board, door) that is closest to the window based on the shortest distance between their closest points. Choose the nearest instance if several exist.
A. Bookcase
B. Chair
C. Board
D. Door

---

Figure 23: Examples of VSI-590K (Annotated Simulated Video (Frame)).

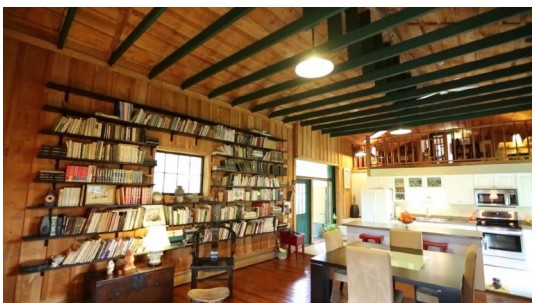

Object Counting (Relative)

If counted, would chairs be fewer than, more than, or equal in number to tables?
A. Fewer
B. More
C. Equal

Relative Direction (Camera Perspective)

Through the camera's lens, is the sink captured on the left or right part of the scene?
A. Right
B. Left

Figure 24: Examples of VSI-590K (Unannotated Real Video (Frame)).

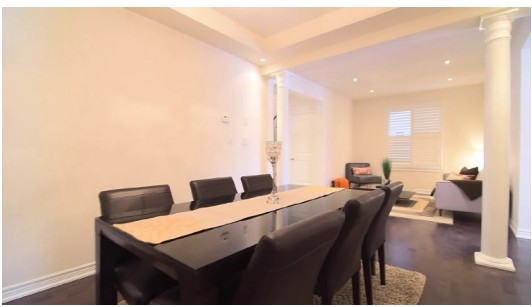

Object Counting (Absolute)

What would be the count if you tallied all the chairs?
Answer: 6

Relative Distance (Camera Perspective)

In terms of proximity to the camera, which is closer: a table or a sofa?
A. Table
B. Sofa

Figure 25: Examples of VSI-590K (Unannotated Real Video (Frame)).

