# OpenReview forum: "Cambrian-S: Towards Spatial Supersensing in Video"
_ICLR.cc/2026/Conference — ICLR 2026 Poster_

### Official Review · Reviewer_Y21j · 2025-10-27

**Soundness:** 3
**Presentation:** 2
**Contribution:** 3
**Rating:** 6
**Confidence:** 3

**Summary:**

The paper proposes spatial supersensing as a framework for video understanding with four levels: semantic perception, streaming event cognition, implicit 3D spatial cognition, and predictive world modeling. It contributes: (1) VSI-SUPER, a benchmark for continual spatial sensing with two tasks (VSO for long-horizon recall, VSC for continual counting), (2) VSI-590K, a large-scale spatial instruction-tuning dataset combining real, simulated, and pseudo-annotated data, (3) Cambrian-S, a family of video MLLMs achieving SOTA on spatial tasks, and (4) a predictive sensing approach using latent frame prediction for memory management and event segmentation.

**Strengths:**

S1: It correctly and timely identifies that current video benchmarks focus on semantic perception while neglecting spatial/temporal reasoning.

S2: It creates a Comprehensive dataset, VSI-590K which combines real annotated, simulated, and pseudo-annotated data with clear methodology.

S3: Strong Baseline as Cambrian-S and novel and creative application of prediction error/"surprise" for both memory compression and event segmentation.

**Weaknesses:**

W1: Claims "data alone isn't enough" doesn't have proper justification. VSI-590K focuses on spatial cognition, not streaming/continual tasks. So, the failure on VSI-SUPER doesn't prove data is insufficient, it seems to show that they didn't collect data for that problem.

W2: There are a lot of missing justifications for core design choices, for example:
(a) Why is cosine distance a valid surprise metric? Any intuition or citation?
(b) Lines 214-215, 319, 364 reference predictive sensing, surprise signals and other neuroscience related terms without proper citations, which might be difficult for the intended audience of the paper to understand.
(c) Why 2-layer MLP for prediction?

W3: Predictive sensing is only tested on two specialized tasks (VSO, VSC). The generalizability to other spatial reasoning scenarios is not very clear.

W4. The pipeline for generating pseudo-annotations (Fig. 5) could introduce systematic biases. No analysis of annotation quality is provided.

W5. No human baseline on the proposed benchmark.

---

Minor:

1. Writing is not very clear and coherent.

2. Typo: line 317, add space between fixed-context and Cambrian-S.

3. Line 1024: GPT, which version?

4. Missing related works: I see some other works which seems related to this and try to understand spatial and/or temporal reasoning of MLLMs, long video understanding and world models. I think authors should add some of these at least in the related work section [1, 2, 3, 4, 5, 6, ...] .  I know that the field is very dynamic and it's not possible to add all the papers, so they should consider adding these and other papers they can find as a minor suggestion.

---

1. Lu, Yiren, et al. "Bard-gs: Blur-aware reconstruction of dynamic scenes via gaussian splatting." Proceedings of the Computer Vision and Pattern Recognition Conference. 2025.

2. Kang, Bingyi, et al. "How far is video generation from world model: A physical law perspective." arXiv preprint arXiv:2411.02385 (2024).

3. Shen, Xiaoqian, et al. "Longvu: Spatiotemporal adaptive compression for long video-language understanding." arXiv preprint arXiv:2410.17434 (2024).

4. Upadhyay, Ujjwal, et al. "Time Blindness: Why Video-Language Models Can't See What Humans Can?." arXiv preprint arXiv:2505.24867 (2025).

5. Song, Chan Hee, et al. "Robospatial: Teaching spatial understanding to 2d and 3d vision-language models for robotics." Proceedings of the Computer Vision and Pattern Recognition Conference. 2025.

6. Lin, Yueqian, et al. "Hippomm: Hippocampal-inspired multimodal memory for long audiovisual event understanding." arXiv preprint arXiv:2504.10739 (2025).

**Questions:**

Q1. Why is cosine distance a valid surprise metric?

Q2. Why the complete failure on longer videos? Table 2 shows 0% on VSC for 120-minute videos. Is this a memory issue, accumulation of errors, or fundamental limitation?

Q3. What about streaming video at higher frame rates? 1 FPS sampling may miss important events.

Q4. Is it possible to get the human accuracy even on some sample of the data?

---

> ### Author Response · Authors · 2025-11-21
> **Official Comment by Authors (P1)**
>
> > W1: Claims "data alone isn't enough" doesn't have proper justification. VSI-590K is not collected for VSI-SUPER task.
> >
>
> We value your perspective on the impact of training data on VSI-SUPER performance. To investigate this, we curated 2,000 videos, each 10 minutes in duration, by concatenating shorter clips (following the same pipeline as VSI-SUPER) and generated 10k QA pairs focused on object counting. These data were mixed into Stage 4 training, after which we evaluated performance on VSC tasks.
>
> As the results bellow show, although adding 10-minute video data significantly boosts performance on the VSC 10-minute subset, **it fails to generalize to the 30-minute subset, and performance on the 60-minute subset remains zero**. This supports our claim in Line 216: VSC requires a generalization of counting behavior that is difficult for the current MLLM paradigm to achieve, as it is impractical to sample data covering all possible video durations and target counts.
>
> |  | 10 mins | 30 mins | 60 mins | 120 mins |
> | --- | --- | --- | --- | --- |
> | Cambrian-S | 16.0 | 0.0 | 0.0 | 0.0 |
> | Cambrian-S (w/ 10K VSC 10mins data) | 58.0 | 11.0 | 0.0 | 0.0 |
>
> > W2 & Q1: Missing justifications: (1) why is cosine distance a valid surprise metric? (2) Lines 214-215, 319, 364 lack proper citations. (3) Why 2-layer MLP for prediction?
> >
>
> Thank you for raising these question. Bellows are our responses:
>
> (1) The reasons to use cosine distance as surprise metric are two-fold:
>
> - First, during training, we apply MSE and cosine loss to optimize our next latent frame prediction objective. We chose cosine distance instead of MSE because MSE is unbounded and difficult to normalize, often exhibiting high variance across samples.
> - Second, cosine distance effectively measures feature similarity and is widely adopted in memory frameworks such as MovieChat.
>
> (2) We thank the reviewer for pointing this out. In response, we have updated the manuscript to include the necessary related work, which improves the overall clarity and context of the paper.
>
> (3) During our augmented stage 4 finetuning (detailed in Sec. 4.1), we train the language model and vision-language connector end-to-end. Language model contribute to 7B learnable parameters during training, which ensures sufficient parameters are optimized by next latent frame prediction objective. However, due the the mismatch between the output feature dimension of the language model and SigLIP2 vision encoder, so we introduce a light weight module (2-layer MLP) to align the feature dimension while introducing negligible new parameters.

---

> > ### Author Response · Authors · 2025-11-21
> > **Official Comment by Authors (P2)**
> >
> > > W3: Predictive sensing is only tested on VSO and VSC, the generalizability to other spatial reasoning scenarios is not clear.
> > >
> >
> > Thanks for your constructive feedback. It is noteworthy that the predictive sensing paradigm is designed specifically for **continuous and unbounded** **visual streams.** It employs specific techniques like **window attentions** and **frame compressions** (see Figure 8), which are unnecessary and suboptimal for short and offline videos.
> >
> > However, we would like to follow your suggestion and evaluate our predictive sensing paradigm on several existing video understanding benchmarks, and compare with previous long-video understanding paradigm (i.e., MovieChat and Flash-VStream).
> >
> > Specifically, we adapt our predictive-sensing-driven memory framework to VSI-Bench, VideoMME, and EgoSchema, using predictive error as a guidance for memory compression and consolidation (same as for VSO). As the table bellow shows, compared to existing long video paradigm, our method exhibits nontrivial advantages.
> >
> > | Method | Video Sampling | VSI-Bench | Video-MME | EgoSchema |
> > | --- | --- | --- | --- | --- |
> > | Naive Inference | 1 FPS | 65.3 | OOM | 76.8 |
> > | With Memory and Predictive sensing | 1 FPS | 64.7 | 61.3 | 75.8 |
> > | MovieChat | 1 FPS | 53.3 | 59.4 | 74.7 |
> > | Flash-VStream | 1 FPS | 52.1 | 55.4 | 73.3 |
> >
> > > W4: The pipeline for generating pseudo-annotations (Fig. 5) could introduce systematic biases. No analysis of annotation quality is provided.
> > >
> >
> > We sincerely appreciate your suggestion to conduct a validation analysis on our pseudo-annotation pipeline. To achieve this, we randomly sampled 500 questions from the pipeline-generated data and asked human annotators to answer them. By comparing the generated annotations against the human ground truth, we measured the accuracy across different tasks (shown in the table below). In general, the pipeline achieves 64.2% Mean Relative Accuracy (MRA) on the absolute count task and between 60% and 90% accuracy on other tasks. These results align with our observations in Table 6, where real and simulated videos yielded better performance than pseudo-annotated images, likely due to inevitable systematic errors in the annotation process. Consequently, we minimized the impact of these errors by limiting pseudo-annotated data to a small fraction (38K out of 590K) of the training set. See Table 5 in Appendix for reference.
> >
> > | Question Type | Metrics | Verification Res. |
> > | --- | --- | --- |
> > | Absolute Count | MRA | 64.2 |
> > | Relative Count | Acc | 65.0 |
> > | Relative Direction (Camera) | Acc | 83.8 |
> > | Relative Direction (Object) | Acc | 64.0 |
> > | Relative Distance (Camera) | Acc | 86.3 |
> > | Relative Distance (Camera) | Acc | 78.6 |
> >
> > We also included these results in our revision, please refer to Appendix Table 9.
> >
> > > W5 & Q4: Human performance on VSI-SUPER.
> > >
> >
> > We appreciate this suggestion and agree that establishing a human baseline is crucial. In response, we evaluated human performance on both the VSC (10-minute) and VSO (60-minute) versions. Specifically, we recruited 10 human annotators to complete the tasks independently, without providing specific training. The results below indicate that humans achieve near-perfect performance on VSO (95.2% accuracy) and strong performance on VSC (76.5 MRA), both of which are much better than today's frontier MLLMs.
> >
> > |  | Metric | Human Performance | Gemini-2.5-Flash | Cambrian-S-7B (w/o. LFP) |
> > | --- | --- | --- | --- | --- |
> > | VSC (10mins) | MRA | 76.5 | 31.2 | 16.0 |
> > | VSO (60mins) | Acc | 95.2 | 34.7 | 0.0 |
> >
> > These results are also included in our revision (Appendix Sec. C.3)
> >
> > > Minor 1 & 2: Writing is not very clear and coherent. Typo in Line 317.
> > >
> >
> > A: Thank you for your careful review. For the typo, we have already updated our manuscripts accordingly. We are also actively reorganizing and polishing our work, and will update in the next revision.
> >
> > > Minor 3: GPT version of Line 1024
> > >
> >
> > A: We used GPT-4o to rewrite and augment the prompt. We also updated it in our manuscript for clarity.
> >
> > > Minor 4: Missing related work.
> > >
> >
> > A: Thank you for sharing these related works with us. We have added these papers to our related work (Section B in Appendix) in this revision. Additionally, we have added a dedicated paragraph to discuss world model related work.

---

> ### Author Response · Authors · 2025-11-21
> **Official Comment by Authors (P3)**
>
> > Q2: Why the complete failure on longer video? Table 2 shows 0% on VSC for 120-minute videos. Is this a memory issue, accumulation of errors, or fundamental limitation?
> >
>
> Thanks for your constructive question. This complete failure is attributed to both context length issue and fundamental limitation of current MLLM.
>
> **First**, the Cambrian-S model has a context length of **128K.** Since we use 64 tokens for each frame and sample at 1 FPS, the model runs out of context for setups longer than **30 minutes**. This is why the Cambrian-S models score **0%** on the VSO and VSC tasks for the **60, 120, and 240-minute** video lengths. We include such clarification in the revised paper.
>
> **Second**, even when the context is long enough, the VSC task is still **fundamentally difficult**. For example, as shown in Figure 13, **Gemini-1.5-Flash** still scores **0%** on VSC for the 60-minute task, even though it is within its context limit.
>
> The difficulty of VSC stems from the following three perspective:
>
> - **Unbounded Visual Stream Understanding**. To achieve good performance, MLLMs need to be able to perceive hours-long video even arbitrarily long video. However, current MLLMs are limited by their context length and can hardly achieve this.
> - **Spatial Reasoning**. Models need to count objects in 3D space. This requires not only frame-level recognition but also the ability to identify the same instance across different frames and perspectives to avoid double-counting.
> - **The ability to generalize to any number for counting**. Models need to be able to generalize to arbitrary counts. As noted in the paper, VSI-SUPER simulates arbitrarily long videos by concatenating clips; as the video length increases, the total number of objects increases significantly. While this scaling is trivial for humans, it poses a substantial challenge for current MLLMs. As shown in Fig. 15, while the ground truth object count scales to over 200 or 300, existing models struggle to count beyond a certain range (e.g., Gemini-1.5-Flash’s prediction plateaus near 50). We hypothesize that this occurs because models are trained on numbers within a restricted range and cannot generalize beyond that distribution. However, simply adding more data cannot solve this problem due to the inherent infinite vocabulary size of the counting task.
>
> These three capabilities required by VSC pose great challenge on current MLLMs and results in the poor performance.
>
> > Q3: What about streaming video at higher frame rates? 1 FPS sampling may miss important events.
> >
>
> We appreciate this insight and agree that the ultimate goal is for MLLMs to perceive the world at a much higher frequency (e.g., 24 FPS, similar to human perception). However, most current MLLMs adopt 1 FPS or uniform sampling as a necessary compromise due to the prohibitive training and inference costs associated with higher sampling rates.
> For example, sampling a 10-minute video at 24 FPS (with 64 tokens per frame) would yield approximately 1 million tokens, which exceeds the context length of most existing models. Given these constraints, even leading proprietary MLLMs typically use 1 FPS sampling to accommodate hour-long videos. We have added a discussion regarding this trade-off to our new 'Limitations and Future Work' section (see Appendix Sec. I).

---

> > ### Comment · Reviewer_Y21j · 2025-11-25
> >
> > I thank the authors for the thorough rebuttal. It resolved most of my concerns.  I encourage the authors to discuss the generalizability results (W3 response) more prominently in the final version, as the mixed performance on standard benchmarks is important context. I will increase my score and hope the authors will improve readability in the final version.

---

> ### Author Response · Authors · 2025-11-27
> **Official Comment by Authors**
>
> We sincerely thank you for your response and appreciate you mentioning will increase the score. We will include the new experiments and discussions mentioned in our response to W3, and we will polish the paper's structure and organization in the final revision.

---

### Official Review · Reviewer_32fi · 2025-10-30

**Soundness:** 3
**Presentation:** 3
**Contribution:** 3
**Rating:** 4
**Confidence:** 4

**Summary:**

In this paper, the authors try to investigate the problem of long-stream spatial cognition issue when applying very long spatial video input, which raises difficulties of long-context and memory to handle well with spatial reasoning, planning, or continual comprehensions, etc. This paper then proposes a very hard dataset to comprehensively test the powerful multimodal image/video MLLMs, and finds that these powerful models still face serious bottlenecks to resolve the long-duration streaming video to maintain the useful and informative contexts or memory. To evaluate this ability, this paper proposes two benchmarks, so-called VSI-Super from both Long-horizon Spatial Observation and Recall and Continual Counting under Changing Viewpoints and Scenes, to evaluate MLLMs to observe long spatiotemporal videos and recall the specific locations of an unusual object in the correct order of its appearance, and continuous unique object counting in long-form spatial videos. Due to the limited resources and unbounded consumption, even the very powerful MLLM cannot solve these tasks well. A new dataset, VSI-590K, is then proposed to upgrade the CAMBRIAN-1 to achieve more powerful spatially-grounded models, ranging from 0.5B to 7B scales. This paper utilizes a concept, predictive sensing, to train the model to align with the latent feature of the next frame based on the current frame, which is then used to quantify the frames' context memory so as to handle the very long spatial video input by simulating the human's memory. After that, this paper obtains improved performances across various spatial cognition benchmarks.

**Strengths:**

1. The huge effort to collect and curate the VSI-Super benchmark and VSI-590K dataset demonstrates the great workload of this paper, which I think this can boost the spatial intelligence community if released with high-quality.

2. The so-called predictive sensing, which is modeled by next frame prediction, sounds like a reasonable way to maintain the history memory context for the scenarios that go smoothly and do not change the scene or even entities drastically.

3. Organizing the next latent frame error as a signal to decide the memory saving cost for the long-horizon context sounds interesting, though this seems very sensitive and vulnerable for the complex scene videos.

4. The experimental attempts are extensive.

**Weaknesses:**

1. The VSO task, which requires MLLMs to observe long spatiotemporal videos and recall the specific locations of an unusual object in the correct order of its appearance, sounds very similar to Needle In A Video Haystack and the common spatial perception task, which requires object appearance order. Can the authors explain and demonstrate the main differences and also the motivations?

2. Regarding the VSC task, does the model need to recognize and count the objects from different instance levels or just the category level? Is this within the very fine-grained object perception?

3. From Figure 5, it is unclear how the paper lifts the 2D image into the 3D space. Can the authors provide more details?

4. Regarding the video scenario, when the event scene changes into another new one, does the model still need to take efforts to store these old history contexts, since our human may try to forget this or not? I notice this paper sets up a fixed window to take the event memory in Fig. 12; however, what if the user asks for a retrieval or comparison with a very old object within the already cleaned memory pool?

5. Regarding the VSO memory design, what are the differences when compared with the SAM2 memory design, Moviechat, or Flash-vstream?

6. Though promising, the so-call supersensing sounds similar with the world model and long-video understanding, can authors provide with better demonstration about the critical differences?

7. The writing organization looks a bit chaos and there is a typo in Line316-317.

**Questions:**

Hope the author can explain and demonstrate well the question above. I will consider raising my score if the authors rebuttal well.

**Details Of Ethics Concerns:**

No.

---

> ### Author Response · Authors · 2025-11-21
> **Official Comment by Authors (P1)**
>
> > W1: Differences between VSO and Needle In a Video Haystack benchmark, and the motivation of VSO.
> >
>
> A. Thanks for pointing out the need for a more detailed discussion and comparison between VSO and other video needle-in-a-haystack benchmarks. We acknowledge that our VSO can be viewed as one type of Video NIAH benchmark. In the following, we distinguish our VSO from existing ones from three perspective:
>
> - **VSO is designed to reduce artifacts.** Existing video NIAH benchmarks like V-NIAH[1] and VideoNIAH[2] insert random images with significant different appearance than other frames into video, which introduce a lot of artifects and bring some trivial solutions (e.g., simply compare the appearance or adjacent frame difference can identify the needles from the frames). To remedy this issue, we asked human annotators to use image editing tools to insert novel object in the original frame, which makes the video looks more coherent and real.
> - **VSO requires not only temporal understanding but also spatial reasoning.** Previous video NIAH benchmarks typically focus on needle information recall which requires temporal understanding and basic semantic understanding. Contraversarily, our VSO asks the order of object's placements, which requires both temporal understanding and spatial reasoning, and poses more challenge for current MLLMs.
> - **Existing video NIAH can be cracked by context lengths, but however, scale up context length only cannot solve VSO yet.** For example, with a long-context base language model, the finetuned MLLM can perform almost perfect on V-NIAH [1]. And on VideoNIAH [2], *Gemini-1.5-Pro can already secure 90.7% accuracy on retrieval task and 72.9% accuracy on ordering task*. However, as Table gemini-2.5-Flash with 1 million tokens context length can only get 34.7 on our VSO (60 mins) task.
>
> ```
> [1] Zhang, Peiyuan, et al. "Long context transfer from language to vision." arXiv preprint arXiv:2406.16852 (2024).
> [2] Zhao, Zijia, et al. "Needle in a video haystack: A scalable synthetic evaluator for video mllms." arXiv preprint arXiv:2406.09367 (2024).
> ```
>
> > W2: Does model need to recognize and count objects from different instance levels or just category level for VSC task? Is this within fine-grained object perception?
> >
>
> A: In our VSC task, models are required to count **the number of distinct instances of a certain category** (e.g., chair) across all spaces in the concatenated long video. To ensure valid evaluation, we intentionally constrained the target objects to be common and sufficiently large to ensure perceptibility. We validate the task's feasibility through human evaluation on a 10-minute subset. Human evaluators achieve 76.5% MRA, significantly outperforming Gemini-2.5-Flash, which reached only 31.2%. We include this human evaluation result in the revision.
>
> > W3: More details about how to lift 2D image into 3D space.
> >
>
> A: Thanks for your question! We explain our pseudo-annotation process for unannotated real videos in detail in the Appendix (Algorithm 1: lines 1102–1132).
>
> Specifically, to transform 2D image into 3D space, we use **VGGT** to extract 3D point map for each image. We then combine this map with the instance masks we created earlier.
> A key step is using an erosion algorithm to clean up the 3D instance masks. This is important because it removes inaccurate 3D points near the edges of objects. Without this step, our estimates for distance, size and direction would be significantly inaccurate.
>
> In addition to transforming 2D images into 3D space, we obtain semantic masks for objects in certain video frames. We do this by:
>
> 1. Sample frames at regular intervals and filter out blurry ones.
> 2. For each retained frame, detect objects using an open-vocabulary detector Grounding-DINO with a predefined list of object categories of interest.
> 3. If the frame contains sufficient objects of interest, we apply SAM2 to generate the final object masks.
>
> We also include these details to the appendix in our revision (please refer to Appendix Sec. D.2).

---

> ### Author Response · Authors · 2025-11-21
> **Official Comment by Authors (P2)**
>
> > W4: Regarding the video scenario, when the event scene changes into another new one, does the model still need to take efforts to store these old history contexts, since our human may try to forget this or not? I notice this paper sets up a fixed window to take the event memory in Fig. 12; however, what if the user asks for a retrieval or comparison with a very old object within the already cleaned memory pool?
> >
>
> A: We appreciate this insightful question. As shown in Fig. 12, upon the appearance of a surprise frame, we use previous memories to compute the count for that specific video chunk and then aggregate the chunk-wise results. In our current implementation, we discard these frame features after answering to maintain a clean working memory.
>
> However, one could also store these memory units in a long-term memory bank (similar to our VSO mechanism, which applies a consolidation function to maintain the memory budget) to facilitate answering related questions in the future.
>
> This question highlights a fundamental dilemma in current representation learning and world modeling: the trade-off between structured semantic representation and lossless representation. Ideally, video understanding requires representations that are both highly structured and capable of capturing every detail from the raw visual signals (lossless). However, current representations fall short of this goal, making it easy to design tasks that challenge MLLMs, like counting numbers of red pixels. As a compromise, we believe the memory bank should prioritize generally meaningful information while discarding likely irrelevant details. Regarding the VSC task, we discard previous features because we have prior knowledge of what is meaningful information (i.e., the number of certain object).
>
> > W5: What's the difference between VSO memory design and SAM2, MovieChat, and Flash-VStream?
> >
>
> We appreciate the suggestion to provide a more comprehensive comparison of our memory design against existing methods. We detail these differences across four key perspectives:
>
> - Basic Memory Unit: Both our VSO and VSC frameworks utilize the LLM KV-Cache of frame features as the basic memory unit. In contrast, SAM2, MovieChat, and Flash-VStream employ ViT-encoded frame features as their basic units.
> - Frame-wise Memory Compression: Our VSO memory compresses frame inputs based on a surprise score threshold. Conversely, SAM2 and MovieChat do not perform frame-wise compression. Flash-VStream compresses per-frame memory using abstract tokens and a learnable semantic attention module.
> - Memory Bank Consolidation: Consolidation is crucial for maintaining a fixed memory budget. In VSO memory, when the token count exceeds the limit, we merge or drop frames based on their surprise scores. SAM2 maintains its budget via a FIFO queue. MovieChat merges adjacent frames with high similarity, whereas Flash-VStream adopts a weighted K-Means clustering algorithm to select the top-K frames.
> - Memory Retrieval: Memory should not only store data but also efficiently retrieve relevant information for problem-solving. Driven by this goal, we calculate the KV-Cache similarity between the question and frames to retrieve the top-K relevant frames, attending only to these during inference. Notably, SAM2, MovieChat, and Flash-VStream lack this specific retrieval design.
>
> We emphasize that the core idea of VSO memory is using “surprise” to guide memory management: high-surprise memories are retained longer and attended to more heavily, while low-surprise memories are compressed or dropped.
>
> We compare our memory design with MovieChat and Flash-VStream on our VSI-SUPER benchmark in Table 17. Below, we provide additional comparisons on VSI-Bench, VideoMME, and Egoschema.
>
> | Method | Video Sampling | VSI-Bench | Video-MME | EgoSchema |
> | --- | --- | --- | --- | --- |
> | Naive Inference | 1 FPS | 65.3 | OOM | 76.8 |
> | With Memory and Predictive sensing | 1 FPS | 64.7 | 61.3 | 75.8 |
> | MovieChat | 1 FPS | 53.3 | 59.4 | 74.7 |
> | Flash-VStream | 1 FPS | 52.1 | 55.4 | 73.3 |
>
> We highly value your question and in response we also updated our manuscript to provide more implementation details on our memory framework (see Appendix Sec. H)

---

> ### Author Response · Authors · 2025-11-21
> **Official Comment by Authors (P3)**
>
> > W6: Differences of supersensing when compared to world model and long video understanding.
> >
>
> A: We appreciate your suggestion to clarify the distinctions between supersensing, world models, and long-video understanding.
>
> - Supersensing: As defined in our paper, supersensing is a fundamental capability intrinsic to multimodal models. We organize this capability into a four-level taxonomy, ranging from semantic perception and streaming event cognition to implicit 3D spatial cognition and predictive world modeling.
> - Long-Video Understanding: In contrast, long-video understanding is a specific task domain. While various independent methods (e.g., tool-use, coarse-to-fine strategies, or divide-and-conquer frameworks) can be designed to address this task, we argue that models possessing "supersensing" capabilities should be naturally able to solve long-video understanding tasks in an efficient and effective manner.
> - World Models: World models are a group of models designed to represent and simulate aspects of the external environment to facilitate generation, planning, or decision-making. While recently a growing body of work has explored the idea of world modeling through self-supervised representation learning, and text- or action-conditioned video generation, in this paper we view the "internal predictive world model" as one of the key aspects of supersensing capability. It serves as the engine that allows a multimodal agent to effectively perceive and understand unbounded visual sensory inputs. To prototype this "predictive sensing," we trained our model with a latent-frame-prediction objective. We conducted two case studies on VSI-SUPER to demonstrate specifically how this predictive world model contributes to the broader capability of supersensing.

---

> ### Author Response · Authors · 2025-11-21
> **Official Comment by Authors (P4)**
>
> > W7: Chaos writing organization. Typos in Line 316.
> >
>
> We sincerely appreciate your careful review. We have corrected the typo and are currently reorganizing the manuscript to improve flow. To strictly adhere to conference policies, we will incorporate these major updates in the final revision.

---

### Official Review · Reviewer_x9om · 2025-11-01

**Soundness:** 3
**Presentation:** 3
**Contribution:** 3
**Rating:** 6
**Confidence:** 4

**Summary:**

This paper frames "spatial supersensing" as a long-term goal for multimodal AI, proposing a four-level taxonomy of capabilities: 1) semantic perception, 2) streaming event cognition, 3) implicit 3D spatial cognition, and 4) predictive world modeling.  The authors argue that current models and benchmarks are stuck at the first two levels, failing to address the challenges of continual, long-horizon spatial reasoning.

To demonstrate this gap, the paper makes three primary contributions. First, it introduces VSI-SUPER, a benchmark with two tasks (VSO for long-horizon recall and VSC for continual counting) specifically designed with arbitrarily long video streams to break the current "brute-force long-context" paradigm. Second, it curates a large-scale spatial instruction-tuning dataset (VSI-590K) and trains a new SOTA model family (Cambrian-S) that excels on existing spatial benchmarks. Third, it shows that this SOTA model still fails on VSI-SUPER, demonstrating a fundamental paradigm gap. The fix is simple: add a latent frame prediction head and use surprise (prediction error) to drive memory compression and event segmentation, which stabilizes accuracy with sequence length.

**Strengths:**

1. Originality and Significance: The paper’s primary strength is its insightful framing. The 4-level taxonomy is a clear and useful way to structure the field's challenges.  The diagnostic audit of existing benchmarks (Fig. 2), which shows many are solvable with text captions, is a solid contribution that validates the need for VSI-SUPER.

2. The task design of VSO is well-grounded.

2. The experimental structure is very effective at proving the paper's story.

**Weaknesses:**

Overall, this is a good work, although with some overclaims.

1. **Limited Task Complexity:** While VSI-SUPER is effective at probing long-horizon memory, the tasks themselves are synthetic and narrow. VSO relies on finding artificially inserted objects, and VSC is a simple counting task (more on "why calling it simple when frontiner models fail later). This is a reasonable first step, but these tasks do not yet capture the full scope of "spatial supersensing," which should arguably involve more complex, emergent reasoning about object interactions, causality, or multi-step agentive plans over time.

2. **Design philosophy for VSC:** I will take an opposite stance on the VSC design and the proposed solution.
  - The VSC benchmark (continual counting) feels co-designed to justify the proposed "surprise-driven **segmentation**" solution. The paper frames this as a failure of long-context models, but simply creating a task that is infeasible for a standard MLLM is not, by itself, evidence of a "paradigm gap." The VSC benchmark seems to be designed for the proposed method rather than the proposed method is designed for the benchmark.
   - If we have the prior that this task requires divide-and-conquer at first, it is easy to come up with other divide-and-conquer methods. For example, (1) a tool-use baseline that first passes the video through a standard shot-detection algorithm and then counts within each segment. Shot-detection algorithms, which detects shot-transition, are very mature and widely used in data preparation pipelines for video generation models. (2) A non-learning baseline that uniformly clips the video into chunks, counts within each, and sums the results (which would likely perform well, barring minor errors at clip boundaries). Without these comparisons, it is unclear if the LFP head is a necessary innovation or just an overly complex solution for this specific task. **The difficulty stems from video length, which is solvable by simple partitioning, not from cognitive complexity that would necessitate a new paradigm**.
    - (more on tool-use) It’s trivial to make some infeasible tasks for current MLLMs. For example, asking an MLLM to count every grain of rice in an image is also infeasible or doing 40-digits arithmetic operations, but this a tool-use problem, not necessarily a fundamental cognitive one.

3. (minor) **Limited Scope of Evaluation**: The entire experimental setup, from the VSI-590K dataset to the VSI-SUPER benchmark, is heavily skewed towards **indoor**, slowly-panned videos (e.g., ScanNet, ProcTHOR).  This leaves the robustness of the "predictive sensing" approach as an open question. It is unclear how the "surprise" signal, trained at 1 FPS, would generalize to more dynamic, real-world scenarios like egocentric video or outdoor driving scenes, which feature fast motion, variable frame rates, and different types of "surprising" events.

**Questions:**

1. In Table 2, why is VSO@10min even worse than VSO@30min?

2. Is prediction error (surprise) measured by patch-averaged errors or some global tokens?

3. Why is predictive error a better measurement than ground-truth error? I have a hard time in understanding this. To me, an educated guess would be that the next-frame prediction is purely trained on in-domain, short-clip data. But during evaluation, the model takes in out-of-domain observation (uncommon objects inserted by editing models or uncommon shot transition from room-to-room) that results in higher errors. Any insights from the authors regarding this question?

4. Is it possible to provide a detailed and holistic compute requirement for evaluating the proposed benchmarks? e.g., how many tokens will be consumed for 10/30/60/120 mins-long videos? how many effective frames in total because of subsampled FPS? how many tokens per frame? what's the GPU requirement for running the evaluations?

---

> ### Author Response · Authors · 2025-11-21
> **Official Comment by Authors (P1)**
>
> > W1: Limited task complexity: VSI-SUPER is synthetic and narrow, cannot capture the full scope of "spatial supersensing", which should involve more complex scenarios.
> >
>
> A: We agree with the reviewer regarding the scope of our benchmark. VSI-SUPER is our **initial exploration** into probing spatial supersensing and we acknowledge that it may not perfectly capture the full complexity of "spatial sensing."
>
> VSI-SUPER was motivated by a diagnostic analysis showing that existing benchmarks mainly focus on semantic understanding or knowledge call. In contrast, VSI-SUPER was designed to specifically challenge frontier models and methods through its **spatial focus** and **arbitrarily long sequences,** which makes it unique among current video QA benchmarks.
>
> While the data is **synthetic** due to the **concatenation and image editing**, and the use of indoor room scans limits its scope, we still believe VSI-SUPER is a **valuable starting point** for this research. Tasks like "long-term spatial observation and recall" and "continual counting under changing viewpoints" have clear real-world applications. Also, the indoor setting is already beneficial for applications like **AR/VR** and **robotics.**
>
> We thank the reviewer for this important point. In response, we have added a section to the revised manuscript discussing these limitations and outlining potential future directions (see Appendix Sec. I).

---

> ### Author Response · Authors · 2025-11-21
> **Official Comment by Authors (P2)**
>
> > W2: Design philosophy for VSC. (1) VSC benchmark feels co-designed for the method. (2) With divide-and-conquer prior, it is easy to design other divide-and-conquer methods. (3) The difficulty stems from video length, not from cognitive complexity that would necessitate a new paradigm. (4) It's easy to make some infeasible tasks for MLLMs. VSC is more like a tool-use problem, not necessarity a fundamental cognitive one.
> >
>
> A: We thank the reviewer for the constructive discussion. We hope to address your specific concerns below:
>
> (1) We want to clarify that VSC was developed *before* we devised a predictive sensing solution for it. We valued the VSC task because:
>
> (i) It aligns with real-world application requirements (a real problem).
>
> (ii) It demands several fundamental capabilities that pose significant challenges for existing models: spatial reasoning, infinitely-long visual stream understanding, and the ability to generalize to any number for counting. We will detail these three capabilities later.
>
> (2) We respectfully argue that VSC is not design for divide-and-conquer methods, although we provide such a solution, it is neither the only solution nor the optimal solution.
>
> Instead, We believe that VSC is valuable for probing streaming cognition, spatial reasoning and generalization capability. While many divide-and-conquer solutions exist for VSC, our main goal is to **test MLLM's continual sensing ability** with **the synthetic unbounded video input**.  Therefore, our discussion of *surprise-driven continual video segment* serves merely as a case study to illustrate how predictive sensing can facilitate video understanding in continuous visual streams.
>
> (3) The difficulty of VSC stems not only from video duration but also from strict requirements for spatial reasoning and OOD generalizability. An example is, Gemini-1.5 pro already achieved 77.4\% accuracy on VideoMME Long (30-60 mins).
>
> - **Spatial Reasoning**: Models must first learn to count objects effectively. This requires not only frame-level recognition but also the ability to identify the same instance across various frames and perspective-taking to avoid double-counting. The poor performance of existing models on VSI-Bench object counting subtasks highlights significant room for improvement.
> - **The ability to generalize to any number for counting**. Models must be able to generalize to arbitrary counts. As noted in the paper, VSI-SUPER simulates arbitrarily long videos by concatenating clips; as the video length increases, the total number of objects increases significantly. While this scaling is trivial for humans, it poses a substantial challenge for current MLLMs. As shown in Fig. 15, while the ground truth object count scales to over 200 or 300, existing models struggle to count beyond a certain range (e.g., Gemini-1.5-Flash’s prediction plateaus near 50). We hypothesize that this occurs because models are trained on numbers within a restricted range and cannot generalize beyond that distribution. However, simply adding more data cannot solve this problem due to the inherent infinite vocabulary size of the counting task.
>
> Together with the capability to perceive and understand unbounded visual streams, all these three capabilities are all fundamental limitations in current MLLM paradigm.
>
> (4) We agree that designing "impossible" tasks for MLLMs is trivial; therefore, **we prioritized practicality to avoid this pitfall** during benchmark design. We constructed VSO using image editing tools and VSC using real-world room scans, rather than synthetic data, to minimize artifacts and maximize realism in our best.
>
> While VSI-SUPER is a simplified environment, we believe it partially reflects real-world requirements on VR/AR and robotics. Furthermore, while we agree that VSC could be approached as a tool-use problem, as aforementioned, we hope to use it as a testbed to evaluate MLLMs' visual spatial counting capability instead of tool-use capability.

---

> > ### Author Response · Authors · 2025-11-21
> > **Official Comment by Authors (P3)**
> >
> > > Q1: In Tab.2, why is VSO 10 mins even worse than VSO 30 mins?
> > >
> >
> > A: We appreciate your careful review. After submission, we noticed a randomization issue where the edited frames in VSO were disproportionately clustered. This unintentional bias made the results kind of weird, like 10 mins seems to be harder than 30 mins that you observed. To remedy this, we regenerated all VSO videos with improved randomization for the placement of edited segments. We have updated Table 2 in this revision to reflect the corrected data and results. We also provide the updated results for Figure 9 bellow. Due to rebuttal policy constraints, we will update the full suite of VSO-related results in the final version of the paper.
> >
> > |                           | VSO (10mins) | VSO (30 mins) | VSO (60 mins) | VSO (120 mins) | VSO (240 mins) |
> > |---------------------------|--------------|---------------|---------------|----------------|----------------|
> > | Gemini-2.5-Flash          | 90.0         | 81.7          | 41.6          | 0.0            | 0.0            |
> > | Gemini-2.0-Flash          | 43.3         | 40.0          | 28.3          | 21.6           | 20.0           |
> > | Cambrian-S (w/o. Memory)  | 33.3         | 28.3          | 0.0           | 0.0            | 0.0            |
> > | Cambrian-S (w/. Memory)   | 45.0         | 41.7          | 40.0          | 40.0           | 40.0           |
> >
> > > Q2: Is prediction error measured by patch-averaged errors or some global tokens?
> > >
> >
> > A: In our implementation, prediction error is implemented as the patch-averaged cosine distance between the predicted and ground-truth next frame features. We have updated the manuscript to improve clarity on this point (see Sec. 4.1).
> >
> > > Q3: Why is predictive error a better measurement than ground-truth error? Is it because the model is trained on in-domain short-clip data and out-of-domain observation during test time will result in higher errors?
> > >
> >
> > A: We agree with your opinion that out-of-domain observations typically lead to higher prediction errors.
> >
> > However, we wish to provide an additional perspective. During next-latent-frame-prediction training, the model not only acquires knowledge about the environment but also learns to **predict future states based on past observations**. “Predictive sensing” can leverage this capability to quantify redundancy in video streams.
> >
> > For example, indoor 3D scanning videos often contain repetitive passes to ensure enough information for complete reconstruction. This high level of redundancy poses a significant challenge to Video MLLMs. In this context, predictive error serves as **a continuous measure of redundancy** relative to the model's internal state, whereas simple adjacent frame feature distance captures only **local**, **pairwise** redundancy. By leveraging prediction error to create a more compact video representation, we can significantly benefit downstream understanding tasks, particularly in infinite-long video scenarios.

---

> > > ### Comment · Reviewer_x9om · 2025-11-24
> > >
> > > For Q3, the rebuttal doesn't address my question. I still do not understand why predictive error is better than ground-truth error in quantifying surprise.

---

> > > > ### Author Response · Authors · 2025-11-25
> > > > **Official Comment by Authors**
> > > >
> > > > We sincerely thank you for your response. Regarding your question, we would like to respectfully ask for clarification on the definition of "ground-truth" error.
> > > >
> > > > - If "ground-truth" error refers to adjacent frame feature difference, as stated in our previous response, the advantage of predictive world modeling lies in two key areas:
> > > >     1. **Predictive world modeling measures novelty or surprise over a longer horizon than adjacent frame difference.** Taking a simple case as an example: if two different frames repeat in a loop, one should be able to quickly get familiar with the pattern, resulting in no surprise. However, adjacent frame difference only measures local changes between two frames, which would lead to a high surprise score for every frame in such a case. And here is a more practical case: If the video includes multiple traverse in a room, the second traverse should be less surprise because the model can see the room's content in the first pass. But adjacent frame difference cannot capture this longer-horizon contextual information.
> > > >     2. During self-supervised tuning, the model learns in-distribution knowledge about the room scan videos. Consequently, the appearance of a novel object correctly leads to a high surprise score."
> > > > - If "ground-truth" error refers to the actual timestamps of novel object appearances or scene transitions. Then, there is no **ground-truth error** at all. We wish to emphasize the following:
> > > >     - We identify these specific moments based on their **relatively higher surprise level** compared to other moments, but there is no predetermined, absolute "ground-truth" surprise level that they should match.
> > > >     - We do not claim that predictive error provides superior performance. As illustrated in Figure 13, Cambrian-S w/ GT Seg consistently outperforms Cambrian-S w/ Surprise Seg across all video durations.

---

> > > > > ### Comment · Reviewer_x9om · 2025-11-25
> > > > >
> > > > > I thank the authors for the follow-up and apologize for the ambiguity in my earlier comment. To clarify, by “ground-truth error” I am referring to what you call “Adj. Frame Vision Feature Diff. as Surprise” in Figure 14 (your first point).
> > > > >
> > > > > Your new response gives two intuitive arguments:
> > > > >
> > > > > (1) That prediction error can, in principle, capture longer-horizon familiarity (e.g., repeated traversals becoming less surprising over time), whereas adjacent-frame difference is local.
> > > > >
> > > > > (2) That training on in-distribution room scans makes novel objects naturally yield high prediction error.
> > > > >
> > > > > These are plausible stories, but **they remain purely anecdotal without any empirical support**. To be concrete: I am not asking for a philosophical justification, but for direct evidence. For example, we at least need a plot of prediction error vs. adjacent-frame feature difference as a function of context length / number of previous frames, on a controlled scenario such as:
> > > > >   - a looping sequence (ABABAB…), or
> > > > >   - repeated room traversals, as in your motivating example.
> > > > >
> > > > > In these settings, you could show that prediction error decays over repeated passes while adjacent-frame feature difference stays high, as you claim. Without this type of empirical analysis, the current explanation is just a plausible argument, not a demonstrated underlying mechanism of the actual behavior of your models. What if these models just learn some very simple shortcuts that can hack the proposed benchmarks?

---

> > > > > > ### Author Response · Authors · 2025-11-27
> > > > > > **Official Comment by Authors**
> > > > > >
> > > > > > > The claim of "Paradigm gap" and "data alone is not enough", and the neccessity of predictive sensing.
> > > > > >
> > > > > > - We acknowledge that predictive sensing is not strictly necessary for VSC or VSO when viewed solely through a problem-solving lens. However, our research departs from this engineering-centric perspective. Rather, we focus on a cognitive question: What makes human memory (as a complex multimodal system) so robust? **Humans possess the intrinsic ability to recall distant episodic memories (maybe 10 years ago) without external tools** (such as photos or written records), a feat that current video MLLMs cannot replicate without relying on tool-calling mechanisms. We also respectfully raise the following questions for discussion:
> > > > > > * Imagine a MLLM serving as an assistant in AI glasses or the brain of a home robot, how could it answer a question about an event that occurred **a week, a month, or a year ago** by pushing current paradigm?
> > > > > > * Let's ground this in current technical specifications, assuming a task similar to VSO or VSC. We can adopt the configuration of a state-of-the-art video MLLM as a default setup: 1FPS sampling and 256 tokens per frame. In this regard, 1 hour video generates 921K tokens; 1 day video generates 22M tokens; 1 month video generates 663M tokens; and 1 year video needs around 8T tokens. *Given that humans effortlessly retrieve memories across such time scales, this long-term context requirement is an authentic challenge.*
> > > > > > * This discussion immediately leads to severe scaling questions for the current attention-based architecture:
> > > > > > 	* How can we scale up the current paradigm to have **8T context length** for inference?
> > > > > > 	* What is the practical decoding time for a query given the $O(n^2)$ computational complexity of the attention mechanism on such inputs?
> > > > > >     * What is the **training recipe** for such a model?
> > > > > >
> > > > > > For us, it is the absence of feasible solutions to these architectural and training challenges within the current MLLM framework that justifies our claims of a "paradigm gap" and the limitation that "data alone isn't enough" to bridge it.
> > > > > >
> > > > > > Regarding the comparison between Cambrian-S and Gemini-2.5-Flash on VSO performance at 10 and 30 minutes, we believe comparing a public model to a proprietary model without full architectural disclosure is inherently problematic. Meaningful conclusions are difficult to draw when comparing against a proprietary model, as critical implementation details remain unknown.
> > > > > >
> > > > > > **Two key factors makes Cambrain-S 7B underperforms Gemini-2.5-Flash intuitive:**
> > > > > >
> > > > > > 1. Cambrian-S 7B use Qwen-2.5 7B as its base model, which is much weaker and potentially much smaller than Gemini-2.5-Flash.
> > > > > > 2. Cambrian-S 7B exceeds its context window at 10 minutes video (with a 32K context length), while Gemini-2.5-Flash doesn't exceed its context until 120 minutes (with 1M context length).
> > > > > >
> > > > > > We hope to conduct research on Gemini-2.5-Flash, but we cannot. **What the VSO results truly demonstrate is**:
> > > > > >
> > > > > > 1. State-of-the-art MLLMs still experience performance degradation *within their context length* as videos grow longer—from 10 to 60 minutes.
> > > > > > 2. Cambrian-S maintains stable generalization from 10 minutes all the way to 240 minutes.
> > > > > >
> > > > > > We also acknowledge that, the competitive performance of Gemini-2.5-Flash indicates that scaling data and model size and compute can still lead to nontrivial improvements. However, as discussed above, these scaling strategies are impractical for month-long or year-long scenarios. Besides, we will follow the reviewer's suggestion and add all the discussions above in the final revision.

---

> ### Author Response · Authors · 2025-11-21
> **Official Comment by Authors (P4)**
>
> > Q4: Detailed information about the evaluation: (1) number of tokens for 10/30/60/120 mins long-video. (2) number of frames. (3) number of tokens per frame. (4) GPU requirement for running the evaluation.
> >
>
> A: Follow your suggestion, we list the detailed evaluation setups and GPU requirements in the following tables.
>
> The first table details the evaluation requirements for our Cambrian-S model (without predictive-sensing driven memory) across various video durations. Notably, to successfully process a 120-minute video, the model requires 139GB of GPU memory (H200 GPU).
>
> | Video durations (in minutes) | # of tokens/frame | # of frames | # of tokens | GPU Mem. Req. |
> | --- | --- | --- | --- | --- |
> | 10 | 64 | 600 | 38400 | 33 |
> | 30 | 64 | 1800 | 115200 | 39 |
> | 60 | 64 | 3600 | 230400 | 57 |
> | 120 | 64 | 7200 | 460800 | 139 |
>
> The table bellow details the requirement for our Cambrian-S model with predictive-sensing driven memory framework equipped. As the results show, all experiments share similar memory costs, and can be run on L40S GPU with 40GB GPU Mem.
>
> | Video durations (in minutes) | # of tokens/frame | # of frames | # of token | GPU Mem. Req. |
> | --- | --- | --- | --- | --- |
> | 10 | 64 | 600 | Up to 32768 | 19 |
> | 30 | 64 | 1800 | Up to 32768 | 22 |
> | 60 | 64 | 3600 | Up to 32768 | 23 |
> | 120 | 64 | 7200 | Up to 32768 | 24 |
>
> As a comparison, we also detail Gemini-2.5-Flash's inference strategy in the following table. As the table shows, 60 minutes video will result in more than 1M tokens.
>
> | Video durations (in minutes) | # of tokens/frame | # of frames | # of token | GPU Mem. Req. |
> | --- | --- | --- | --- | --- |
> | ~60 | Unknown | ~3600 | ~1048576 | Unknown |

---

> ### Comment · Reviewer_x9om · 2025-11-24
>
> I thank the authors for the detailed rebuttal and additional analyses. However, my core concerns remain and, after reading other reviews and responses, have actually deepened.
>
> I'm particularly concerned about whether "predictive sensing" paradigm is really justified by the current evidence. It seems the proposed paradigm is *neither the only solution nor the optimal solution*. Some components are unnecessary and suboptimal.
>
> 1. Predictive sensing and the “paradigm gap”
>
> The paper repeatedly promotes “predictive sensing” (via the next latent frame prediction head and surprise-driven memory/segmentation) as a necessary new paradigm for supersensing. After rebuttal, I do not see this as justified.
>
> - The authors themselves emphasize that the proposed divide-and-conquer method is “neither the only solution nor the optimal solution” for VSI-SUPER, and that many divide-and-conquer alternatives exist (such as tool-use, uniform splitting, shot detection + per-split QA). This directly weakens the central “paradigm gap” message, because the paper does not demonstrate that the particular predictive-sensing instantiation is needed rather than one of many simpler engineering baselines.
>
> - From the updated VSO results (in the rebuttal response), Gemini-2.5-Flash is substantially stronger than Cambrian-S (with or without memory) at 10 and 30 minutes, and is competitive at 60 minutes; it only fails when the context window is exhausted (out of 1M context window). This strongly suggests that, for the current VSI-SUPER construction, a simple segment-and-conquer wrapper around an existing long-context model (e.g., run Gemini on 10–30 minute segments and aggregate) would likely perform very well. Yet no such baseline is provided.
>
> - The strong claims that “data alone isn’t enough” and that a qualitatively new predictive-sensing paradigm is needed are less justified. What the paper currently shows is that (i) a strong spatially-tuned model still struggles on an extreme long-stream benchmark, and (ii) one particular heuristic (prediction-error-based surprise) helps on that benchmark. That is interesting, but much weaker than the fancy story the paper is pushing (predictive sensing).
>
> 2. Spatial **"reasoning"**
>
> The authors argue that Cambrian-S with VSI-590K + predictive sensing learns "spatial reasoning ability" to track instances across views and avoid double-counting. I'm not fully convinced by the current results. A simple but telling ablation would be:
>
> - Take a T-minute spatial video with some number of chairs K.
> - Duplicate this video N times to form an NT-minute video where the *ground-truth unique count* is still K (because it is the same environment repeated).
> - Evaluate Cambrian-S and a strong baseline such as Gemini-2.5-Flash on this constructed video.
> - There could be other setups like mixing repeated videos with normal (non-duplicated) random videos. These settings are easy to synthesize with controlled difficulty.
>
> If the models have truly learned instance-level spatial reasoning and are not just summing counts per segment, their predictions should remain close to K as N increases. Right now, the paper infers instance-level reasoning largely from performance on VSC itself, which is not very diagnostic.
>
> ---
>
> I find the benchmark and data curation efforts valuable but the entire scope of the proposed benchmark still lies in the work from Yang et al [1]. The central conceptual message that "predictive sensing is a necessary new paradigm for supersensing and that data/context scaling has hit a fundamental wall", is not convincingly demonstrated by the current experiments.
>
> Arbitrarily infinite long tasks will for sure make all models fail unless you do smart compression or divide-and-conquer. Like what authors agree, it's trivial to design tasks infeasible for current VLMs. It's also trivial to design tasks that requires streaming event cognition and a kind of so-called "reasoning" capability. For example,
> - random streaming digits arithmetic operations like addition, multiplication or subtraction.
> - count random string, e.g., for "A BC E D BC CB C ...." (sufficiently long sequences), how many times does "BC" and "CB" appear? (It's hard to justify counting chairs or other objects is a real-world problem).
>
> I would encourage the authors to significantly tone down the “paradigm gap” and “data alone isn’t enough” claims, and strengthen the empirical story with the kinds of baselines and diagnostics discussed above.
>
>
> [1] Yang, Jihan, et al. "Thinking in space: How multimodal large language models see, remember, and recall spaces." Proceedings of the Computer Vision and Pattern Recognition Conference. 2025.

---

> ### Author Response · Authors · 2025-11-27
> **Official Comment by Authors**
>
> > Spatial reasoning and repeat counting experiments
>
> To clarify, **we do not claim that predictive sensing leads to "spatial reasoning."** Rather, we argue that the object counting task of our **VSC benchmark necessitates spatial reasoning** to prevent double counting.
>
> Per your suggestion, we evaluated our Cambrian-S-7B model on the VSI-Bench "object counting" subtasks using video inputs repeated multiple times. The results, detailed in the table below, demonstrate that our model possesses the requisite spatial reasoning capabilities to avoid double counting, as performance remains stable across repetitions.
>
> | Model | VSI-Bench (1x repeat) | VSI-Bench (2x repeat) | VSI-Bench (3x repeat) |
> |-------|-------|-------|-------|
> | Cambrian-S-7B (uniform sample 128 frames) | 73.22 | 73.18 | 73.25 |
>
> We also evaluated our agentic framework on this repeated counting task. We observed that the framework fails to answer correctly in these instances because the surprise-based scene segmentation method erroneously identifies repeated sequences as distinct scenes. **We attribute this limitation specifically to the agentic framework's segmentation approach rather than to the underlying Cambrian-S model**: our latent next-frame prediction model is trained on natural room scan videos rather than concatenated sequences, it is expected that concatenated videos will yield high surprise values and our agentic framework will take them as two unique rooms.
>
> We thank the reviewer for suggesting these experiments and will include all results in the final revision.
>
> > Quantitive comparision of predictive error and adjacent frame feature difference
>
> We sincerely value your opinion that more through experiments should be conducted. Follow your suggestion, we conducted a repeated sequence experiment to study the difference between "predictive error as surprise" and "adjacent frame feature similarity as surprise".
>
> Specifically, we sampled the first two frames from the 288 videos used in VSI-Bench and repeated them 10 times to form 20-frame sequences (pattern: "ABABAB..."). We then fed these sequences into the model to measure surprise scores using both metrics. For the results, please refer to the latest updated paper.
>
> We visualize these surprise scores in Fig. 17. As shown, "adjacent frame feature difference as surprise" yield constant surprise scores. However, "predictive error as surprise" shows a distinct pattern: the surprise scores will initially decrease and then increase. Here, the surprise is majorly affected by two factors: prior observation, which helps reduce surprise, and the out-of-distribution (OOD) input (the model has never seen these repeated two frames during training), which results in surprise increase. Initially, the prior observation does help to decrease surprise. However, as the sequence gets longer, the increasingly severe OOD input lead to a larger overall increase in surprise.
>
> Besides, we also visualize the distribution of when the minimal surprise score appears in Fig. 18 (Left) and find that: the minimal surprise score (with "predictive error as surprise measurement") frequently appears after the second or the third repetition. Since interleaved repetition causes significant video jitter, we conducted a similar experiment repeating the first two frames 10 times using an "ABBAABBA..." pattern. As shown in Fig. 18 (right), given this smoother video sequence, the minimum surprise score occurs later in the sequence.
>
> We sincerely thank the reviewer for suggesting these experiments and insightful discussion and we will properly include these experiments in the final revision.

---

> > ### Comment · Reviewer_x9om · 2025-11-28
> >
> > I thank the authors for the detailed rebuttal and the substantial additional analysis. I appreciate the back-and-forth discussion during the rebuttal period.
> >
> > I fully agree that 8T tokens or any similarly extreme context length is beyond what current or near-future models can handle in a straightforward way. I personally view this challenge more from an architectural view. For example, linear attention, state-space models, or test-time-training–style approaches also offer potential paths toward effectively unbounded context, with fixed state size and the ability to process very long streams. There is a plausible scenario where these architectures, trained on the proposed data, could solve the current benchmarks (e.g., state-space models are known to be strong at vague retrieval but weaker at precise retrieval, while still maintaining a fixed-size state that can in principle handle very long contexts. (e.g., how we recall childhood memory))
> >
> > Still, I appreciate this work from a “think differently” perspective. It proposes a working path that departs from today’s dominant paradigms, and I see value in that, possibly more than in yet another *transformative* story.
> >
> > I also appreciate the new experiments on predictive error versus ground-truth feature difference. The analysis of the distributions and behaviors of these two indicators is interesting and, in my view, will be useful for readers who want to understand why predictive error provides additional gains.
> >
> > I understand this is additional work, but I have a few final suggestions that can be incorporated before **camera-ready** paper:
> > - Instead of repeating only the first two frames from each video 10 times, it would be more compelling to duplicate full videos. Short 2-frame repetition feels somewhat easy; repeating full videos better stresses models in a way that matches real long-context failure modes.
> > - If there will be a project page, it would be very helpful to overlay predictive error and ground-truth feature error curves under the videos and synchronize them with the video. Showing both repeated sequences and original benchmark samples with these error plots would give readers an immediate and intuitive sense of how these signals behave.
> > - If possible, I suggest toning down the “entirely new paradigm” framing and expanding the discussion of pitfalls and limitations of both existing models and your own approach. No method is free of failure modes, and making these explicitly visible is valuable for future work.
> >
> > It is perfectly fine if the proposed method does not solve all aspects of the problem (e.g., it fails on repeated full videos). Being explicit about where it fails is often how follow-up work makes real progress.
> >
> > In summary, the authors have done a remarkable job in addressing my concerns and have clearly put substantial effort into the revisions. While I still find some claims and interpretations debatable, I believe this is a solid and interesting work. I will therefore raise my final score to 8.

---

> > > ### Comment · Reviewer_x9om · 2025-11-28
> > >
> > > However, ICLR has disabled score editing now. I will update it later.

---

> > > > ### Author Response · Authors · 2025-12-03
> > > > **Official Comment by Authors**
> > > >
> > > > We thank you for the previous discussions and are glad to see that our responses have succesfully addressed your concerns.
> > > >
> > > > Regarding the challenge of extremely long inputs, we respectfully maintain a different view. While it can be partially addressed from an architectural perspective, as discussed in our paper, we believe it is fundamentally a world modeling problem: specifically, how to efficiently and effectively model the world (via language or visual interfaces) within a limited memory budget. Many designs in state space models (SSMs) and linear transformers align with this philosophy. For example, the state update mechanisms in SSMs and the delta rule in DeltaNet.
> > > >
> > > > We also highly value your new suggestions for our upcoming revision. Regarding the full video duplicate experiment, as noted previously, our model is trained exclusively on single-room scan videos; therefore, concatenated or repeated sequences deviate significantly from our training distribution. We will explicitly discuss this as a limitation in the final version. We will also follow your suggestion to visualize the surprise scores and tone down the 'new paradigm' claims.
> > > >
> > > > Finally, we sincerely appreciate your constructive suggestions, which have strengthened our submission. Thank you for recognizing our work and for raising the final score.

---

### Official Review · Reviewer_jh5E · 2025-11-01

**Soundness:** 2
**Presentation:** 3
**Contribution:** 3
**Rating:** 6
**Confidence:** 4

**Summary:**

This work presents a new benchmark and a new paradigm for learning multimodal video intelligence incorporated with the capability of predictive sensing using an internal world model. Extensive experiments and analyses verify that previous methods cannot well address this new problem via simply scaling data and compute and the proposed method achieves significant improvement.

**Strengths:**

1.This work discusses an interesting new paradigm for multimodal video modeling called spatial supersensing, which aims to overcome the limitations of previous methods in predictive modeling driven by internal world model.

2.The proposed predictive sensing paradigm seems to be capable of generalizing to various downstream tasks and could be a more advanced version of multimodal intelligence, supported by a lot of experiments and analyses.

**Weaknesses:**

1.Although the authors claimed that the proposed predictive modeling paradigm better helps downstream video understanding tasks, it seems this argument lacks sufficient experimental evidence. What would be the advantage when it comes to downstream generalizatioin comparing the proposed predictive paradigm and previous paradigms?

2.Another concern is that, from my personal understanding, the proposed framework utilizes the "error" between the predicted next frame latent and the ground-truth next frame latent as a predictive signal. However, as we know that not all future frames are predictable, i.e., some errors are high because of those next frames are unpredictable but some errors are high because the model cannot make accurate predictions. How do the method handle these situations? Will the surprise measurement still make sense when the model comes across some unpredictable frames? And how would such error be utilized for specific video understanding goals?

**Questions:**

Please refer to the weaknesses.

**Details Of Ethics Concerns:**

N/A.

---

> ### Author Response · Authors · 2025-11-21
> **Official Comment by Authors**
>
> > W1: Although the authors claimed that the proposed predictive modeling paradigm better helps downstream video understanding tasks, it seems this argument lacks sufficient experimental evidence. What would be the advantage when it comes to downstream generalizatioin comparing the proposed predictive paradigm and previous paradigms?
> >
>
> A: Thanks for your constructive feedbacks on the generalization of predictive sensing paradigm. First of all, we want to highlight a key point: our diagnostic tests (in Section 2) show that most current video benchmarks knowledge recall and can hardly reflect the unbounded nature of visual sensory inputs.
>
> Our predictive sensing paradigm, however, is designed specifically for these **continuous and unbounded** **visual streams**. To handle such unbounded setting, we use specific techniques like **window attentions** and **frame compressions** (see Figure 8), which is unnecessary and suboptimal for short and offline videos.
>
> However, we still would like to follow your suggestion and evaluate our predictive sensing paradigm on several existing video understanding benchmarks, and compare with previous long-video understanding paradigm (i.e., MovieChat and Flash-VStream).
>
> Specifically, we adapt our predictive-sensing-driven memory framework to VSI-Bench, VideoMME, and EgoSchema, using predictive error as a guidance for memory compression and consolidation (same as for VSO). As the table bellow shows, compared to existing long video paradigm, our method exhibits nontrivial advantages.
>
> | Method | Video Sampling | VSI-Bench | Video-MME | EgoSchema |
> | --- | --- | --- | --- | --- |
> | Naive Inference | 1 FPS | 65.3 | OOM | 76.8 |
> | With Memory and Predictive sensing | 1 FPS | 64.7 | 61.3 | 75.8 |
> | MovieChat | 1 FPS | 53.3 | 59.4 | 74.7 |
> | Flash-VStream | 1 FPS | 52.1 | 55.4 | 73.3 |
>
> > W2: Not all future frames are predictable. Sometimes errors are high because unpredictable frames and sometimes errors are high because model cannot make accurate predictions. How do the method handle these situation? Will the surprise still make sense when the model comes across some unpredictable frames? How such error be utilized for specific video understanding goals?
> >
>
> A: We appreciate and agree your perspective that predictive errors can arise from two primary sources: model inaccuracies and the inherent unpredictability of the frames themselves.
>
> As stated in our paper, our motivation for predictive sensing is that a model should learn an internal representation of the world, so it can actively predict future sensory inputs.
>
> In our implementation, during training on 1FPS sampled indoor scene data, the model is supervised to learn to:
>
> (i) predict the next frame within the latent space, and
>
> (ii) acquire basic structural knowledge of indoor scene videos.
>
> During evaluation, the predictive error serves as a guide for handling unbounded, continuous video efficiently. For example, we could compress less surprising frames more heavily or segmenting video clips based on how surprising they are.
>
> However, we agree that prediction error originates not only from unpredicted frame content (such as novel objects) but also from **the model itself** — specifically, when the model’s internal world model is insufficient to support accurate prediction in certain situations.
>
> For example, consider the analogy of humans entering a new environment (e.g., travel to a new country). Initially, their past experience might not help them accurately estimate what will happen next, which would lead to frequent and significantly "surprises.” However, humans adapt to the environment, learn from feedback, and update their internal world models. Eventually, they become capable to estimate future events again, and less frequently and significantly surprising to the environment.
>
> This capability to quickly adapt the internal world to the outside environments remains largely uncharted in the existing paradigm and leaves significant room for improvement in achieving true "predictive sensing".
>
> And for the unpredictable frame with high prediction error, our VSO memory framework will treat it as a key moment with important information, and store it into the memory bank without any compression and make it more sustainable even when the memory bank exceeds the budget.
>
> We highly value your constructive feedback; in response, we have added a discussion regarding this point to our revision (see Appendix Sec. I).

---

### Author Response · Authors · 2025-11-21
**Official Comment by Authors**

Dear Reviewers, Area Chairs, and Program Chairs,

We sincerely thank you for the time and thoughtful feedback dedicated to our manuscript. We appreciate the reviewers' insightful comments and constructive suggestions, which have helped us significantly improve our paper.

We are encouraged that the reviewers recognized:

- The significance of our supersensing hierarch and detailed diagnostic audit of exisiting benchmarks. (Reviewer x9om and Y21j)
- The effectiveness of our curated large-scale spatial dataset VSI-590K on improving spatial sensing and VSI-SUPER on benchmarking supersensing. (Reviewer 32fi, x9om, and Y21j)
- The competitive performance of Cambrian-S models. (Reviewer x9om and Y21j)
- The novelty of our proposed predictive sensing paradigm (Reviewer jh5E, 32fi, and Y21j)

Based on your valuable feedback, we have revised the manuscript. All major revisions are highlighted in orange for clarity. A summary of the key changes is as follows:

- **More Implementation Details**: We add more details for the unannotated data pipeline (Sec. D.3) and the VSO/VSC memory and agentic frameworks (Sec. H.3).
- **Human Baselines and Data Quality Analysis**: We added human performance on VSI-SUPER (Tab. 4) and quality verification for pseudo-annotated data (Tab. 9).
- **Additional comparisons**: We included comparisons of VSO memory against long-video baselines on VSI-Bench, Video-MME, and EgoSchema (Sec. H.5).
- **Discussion, Limitation, and Future Work**: We added Section I to explicitly address limitations and future work.
- **Typos & Structure**: We corrected typos and are working on major structural reorganization. In accordance with rebuttal policies, major structural changes will be updated in our next revision.
- **More Comprehensive Related Work**: We have added the necessary citations to improve clarity. Additionally, we updated the 'Related Work' section to include other relevant research, including a new paragraph discussing world models (Sec. B).

We provide detailed, point-by-point responses to each reviewer in the individual rebuttal windows.

Thank you again for your constructive feedback, which has been invaluable in improving our paper. We look forward to any further discussion and are happy to address all remaining concerns.

---

### Author Response · Authors · 2025-12-04
**Summary of Clarifications and Evidence Provided During the Rebuttal**

Dear ACs,

We sincerely thank you for your time and thorough consideration of our submission. Below, we provide a concise summary of the factual clarifications and additional evidence included in our rebuttal to address all reviewer concerns.

> Generalizability of predictive sensing and our memory module to other benchmarks. Comparisons between our memory module and exiting methods. (Reviewer jh5E, 32fi, Y21j)

We conducted additional experiments to evaluate our predictive sensing and memory module on new benchmarks and compare them against existing methods. The results demonstrate that our approach consistently outperforms previous methods on VSI-Bench, Video-MME, and EgoSchema.

> Limited task complexity of VSI-SUPER. The difficulty of VSC. (Reviewer x9om, Y21j)

We have expanded our discussion on the limitations of the VSI-SUPER benchmark in the revised paper. We also analyzed the difficulty of VSC tasks from three perspectives: (1) handling arbitrarily long video inputs, (2) performing spatial reasoning, and (3) generalizing to arbitrary counts.

> Comparision of predictive error and adjacent frame feature difference. (Reviewer x9om)

We have provided a thorough analysis distinguishing predictive error from adjacent frame feature difference. Following the reviewer's suggestion, we conducted additional experiments to compare the behavior of 'predictive error as surprise' versus 'adjacent frame feature difference as surprise' on repeated frame sequences. Both the quantitative results and visualizations support our claim that the model learns to predict future frames based on past observations, thereby using historical context to reduce the surprise of incoming frames.

> Spatial reasoning and repeat counting experiments of Cambrian-S. (Reviewer x9om)

To verify whether our Cambrian-S model correctly counts objects and deduplicates repeated instances, we evaluated the Cambrian-S-7B model on the VSI-Bench 'object counting' subtask using repeated video inputs. The results demonstrate that our model possesses the spatial reasoning capabilities necessary to avoid double counting, as performance remains stable across repetitions.

> Detailed evaluation setups and costs. (Reviewer x9om)

We have provided detailed evaluation setups and GPU requirements for various video durations. As the tables demonstrate, our surprise-driven memory framework results in significantly lower and more stable GPU memory consumption.

> More details about how to lift 2D image into 3D space. (Reviewer 32fi)

We have updated our paper to provide a more detailed description (please refer to Appendix Sec. D.2) of how we lift 2D images into 3D space. In summary, we leverage VGGT to extract a 3D point map for each sampled frame and use Grounding-DINO and SAM2 to generate instance masks.

> Finetuning on more VSI-SUPER related data. (Reviewer Y21j)

We curated a new VSC training set comprising 2K videos and 10K QA pairs, with each video lasting approximately 10 minutes. We integrated this data into Stage 4 training and evaluated the model on VSC tasks. As the results show, adding 10-minute counting data significantly boosts performance on the VSC 10-minute subset. However, this improvement does not generalize to the 30-minute or longer duration subsets.

> Quality analysis for pseudo-annotations. (Reviewer Y21j)

We randomly sampled 500 questions from the pipeline-generated data for human verification. Upon comparing the pipeline-generated labels with human annotations, we found that the pipeline achieves 64.2% Mean Relative Accuracy (MRA) on the absolute count task and 60%–90% accuracy on other tasks, demonstrating reliable performance.

> Human performance on VSI-SUPER. (Reviewer Y21j)

We recruited 10 human annotators to complete the VSC (10-minute) and VSO (60-minute) tasks without providing specific training. We found that humans achieved near-perfect performance on VSO (95.2% accuracy) and strong performance on VSC (76.5 MRA), significantly outperforming current frontier MLLMs.

> Writing organizations and typos. Missing related works. (Reviewer 32fi, Y21j)

Following the reviewers' suggestions, we have fixed the typos in our submission and will refine the paper's structure and organization in the final revision. We also expanded the Related Work section to discuss research on world models and included additional references to enhance readability.

All added analyses, tables, and corrections are highlighted in orange in the revised paper. We sincerely appreciate the constructive feedback from all reviewers, and the engagement of Reviewers Y21j and x9om during the discussion period. We are glad that Reviewers x9om and Y21j have acknowledged that our responses successfully addressed their concerns, and we remain happy to answer any further questions.

Best regards,
Authors of ICLR submission 875

---

### Meta-Review · Area_Chair_TBWn · 2026-01-07

**Summary:**

This paper proposes a "spatial supersensing" taxonomy and introduces the VSI-SUPER benchmark to evaluate long-horizon spatial reasoning, arguing for a shift from brute-force context scaling to a predictive sensing approach. The authors demonstrate that their method can generalize to external benchmarks like Video-MME and EgoSchema. While the work offers a valuable perspective, the initial submission faced scrutiny regarding the "paradigm gap" claims and the synthetic nature of the VSC tasks.

**Reviewer Concerns:**

Concerns centered on generalizability (Y21j), the necessity of human baselines (Y21j), and the distinctiveness of the task compared to existing benchmarks (32fi, x9om). The rebuttal effectively addresses scope and generalization by showing performance gains on standard benchmarks (Video-MME, EgoSchema) and providing the requested human baselines (Y21j). Technical clarifications regarding "needle-in-a-haystack" distinctions (32fi) and "repeated frame" ablations (x9om) were also provided. However, x9om noted that the "paradigm gap" claim might be overstated given potential engineering solutions; the authors have agreed to tone down these claims. The limitation in handling full-video repetition remains an open issue but is not viewed as a blocker.

**Reviewer Scores:**

Based on the rebuttal, x9om explicitly raised their score to 8 after the surprise metric analysis. I expect Y21j to increase their score as generalizability concerns were resolved, 32fi to likely raise their score given the technical clarifications, and jh5E to maintain or slightly increase their positive assessment due to the additional downstream task experiments.

---

### Decision · Program_Chairs · 2026-01-26

Accept (Poster)